# Substrate stiffness impacts early biofilm formation by modulating *Pseudomonas aeruginosa* twitching motility

Sofia Gomez[1], Lionel Bureau[1], Karin John[1], Elise-Noëlle Chêne[2], Delphine Débarre[1]*, Sigolene Lecuyer[1,2]*

[1]Université Grenoble Alpes, CNRS, Grenoble, France; [2]Laboratoire de Physique, Université Lyon, ENS de Lyon, Université Claude Bernard, CNRS, Lyon, France

**Abstract** Surface-associated lifestyles dominate in the bacterial world. Large multicellular assemblies, called biofilms, are essential to the survival of bacteria in harsh environments and are closely linked to antibiotic resistance in pathogenic strains. Biofilms stem from the surface colonization of a wide variety of substrates encountered by bacteria, from living tissues to inert materials. Here, we demonstrate experimentally that the promiscuous opportunistic pathogen *Pseudomonas aeruginosa* explores substrates differently based on their rigidity, leading to striking variations in biofilm structure, exopolysaccharides (EPS) distribution, strain mixing during co-colonization and phenotypic expression. Using simple kinetic models, we show that these phenotypes arise through a mechanical interaction between the elasticity of the substrate and the type IV pilus (T4P) machinery, that mediates the surface-based motility called twitching. Together, our findings reveal a new role for substrate softness in the spatial organization of bacteria in complex microenvironments, with far-reaching consequences on efficient biofilm formation.

## Editor's evaluation

This fundamental research significantly enhances our comprehension of the influence of substrate physical properties during the initial stages of biofilm development. By integrating microfluidics, single-cell motility, and modeling, the study provides compelling proof that mechanical interactions between the substrate and Type-IV pili drive these phenomena. This work is likely to attract a wide range of readers interested in micro-communities, their structure, and ecology.

## Introduction

The transition of bacteria from a planktonic to a surface-attached state is of paramount importance in biofilm formation. In consequence, the way bacteria sense and respond to the close proximity of a surface has been the subject of intense scrutiny (*Dufrêne and Persat, 2020*; *Laventie and Jenal, 2020*). This interaction involves different aspects of bacterial motility: swimming toward the surface, but also swarming, gliding or twitching that are used by attached bacteria to explore the surface collectively or individually (*Wadhwa and Berg, 2022*; *Conrad et al., 2011*). Eventually, permanent bacterial adhesion and microcolony structuration may arise, through mechanisms which essential ingredients are known (production of matrix, loss of motility), but in response to cues that remain unclear.

Bacteria are ubiquitous and can successfully colonize a wide range of biological tissues and abiotic surfaces (*Stoodley et al., 2002*; *Mann and Wozniak, 2012*). Different environments often result in different phenotypes for a given microorganism (*Dötsch et al., 2012*; *Cornforth et al., 2018*).

**\*For correspondence:**
delphine.debarre@univ-grenoble-alpes.fr (DD);
sigolene.lecuyer@ens-lyon.fr (SL)

**Competing interest:** The authors declare that no competing interests exist.

However, although chemical signaling has long been known to impact bacterial gene regulation, it remains unclear how the mechanical properties of the encountered surface might impact bacterial behavior (*Persat et al., 2015b*). In this paper, we investigate how the rigidity of a substrate modifies bacterial motility, and by doing so impacts microcolony morphogenesis and early biofilm development.

*Pseudomonas aeruginosa* (PA) is an opportunistic rod-shaped pathogen that contaminates a wide range of substrates, from very soft tissues to rigid implants (*Moradali et al., 2017*; *Chang, 2017*). A particularly gifted and versatile biofilm-former, it is extremely prone to developing antibiotic resistance (*Pang et al., 2019*; *Boucher et al., 2009*). PA has developed an arsenal of techniques to move on surfaces: among them is twitching motility, that allows single bacteria to translocate across surfaces using type IV pili (T4P) (*Maier and Wong, 2015*). T4P are thin protein filaments on the bacterial surface that can extend and contract by assembly and disassembly of the protein subunit PilA. The tip of T4P acts as a promiscuous hook that can grasp most surfaces. Attachment, contraction, detachment and extension cycles propel bacteria (*Merz et al., 2000*; *Maier and Wong, 2015*; *Skerker and Berg, 2001*; *Talà et al., 2019*). This surface motility is important for bacteria to efficiently settle on surfaces, but the exact mechanisms at play are unknown (*O'Toole and Kolter, 1998*; *Leighton et al., 2015*; *Craig et al., 2019*). The function of T4P and the fact that it exerts forces on its environment make it an obvious candidate for surface-sensing mechanisms (*Merz et al., 2000*; *Dufrêne, 2015*; *Sahoo et al., 2016*). Recent results have shown that the polar localization of pili in PA could happen in response to surface-sensing (*Cowles and Gitai, 2010*). Polarly localized pili lead to persistent rather than random displacements, as well as specific effects such as the upstream migration of bacteria submitted to strong flows (*Shen et al., 2012*). Reversal of twitching bacteria is rapidly induced upon meeting obstacles, suggesting a mechanical feedback from T4P (*Kühn et al., 2021*). In addition, PA can exert different types of virulence, from acute attacks to chronic infections (*Furukawa et al., 2006*; *Valentini and Filloux, 2016*), and specific host-pathogen interactions have traditionally been considered as the key players in the regulation of these virulence pathways (*Gellatly and Hancock, 2013*). However, surface-sensing in itself has recently appeared as a potential signal that could trigger the upregulation of virulence-associated genes (*Islam and Krachler, 2016*; *Persat et al., 2015a*; *Siryaporn et al., 2014*). Although the global effect of surface rigidity on bacterial adhesion and biofilm formation has sometimes been addressed (*Saha et al., 2013*; *Song and Ren, 2014*; *Song et al., 2018*), so far how the micromechanical environment experienced by individual bacteria impacts their behavior is still unclear, possibly because of the difficulty to design and control microenvironments that allow for a fine tuning of mechanical properties at the bacterial scale, along with negligible changes of the chemical environment.

In this study, we use a home-designed microfluidic setup to investigate at the single-cell level the influence of substrate rigidity on PA bacteria adhering to an open surface, under controlled flow conditions. We first demonstrate that substrate elasticity strongly impacts early microcolony development. Focusing on single-cell behavior, we then study quantitatively how rigidity modulates bacterial motility and propose a purely mechanistic model to account for our observations. Finally, we demonstrate that this mechanical tuning of the motility explains rigidity-induced changes in early surface colonization: we explore its consequences in terms of microcolony morphology, matrix deposition, strain mixing, and long-term gene expression.

## Results

In order to explore in situ the effect of substrate rigidity on the behavior of adhering bacteria, we have developed an experimental approach to include mechanically well-defined hydrogel pads in a microfluidic channel providing controlled flow conditions and allowing confocal imaging (*Figure 1A*). We use the biocompatible hydrogel polyacrylamide (PAA), which has been extensively used to investigate cell-substrate interaction and mechanotransduction in mammalian cells. By varying the amount of bisacrylamide cross-linker during its preparation, PAA can span a biologically-relevant range of rigidities (from ~1–100 kPa) while keeping a low viscous dissipation. Several pads, with different's modulus values ranging from ~3 to 100 kPa (see *Figure 1—figure supplement 1*, and Materials and methods), are used in each single experiment, and bacteria adhering on PAA and glass surfaces are imaged with high-resolution phase-contrast and fluorescence time-lapse imaging, from very low surface coverage up to the formation of microcolonies (1 frame/min over ~10 hr).

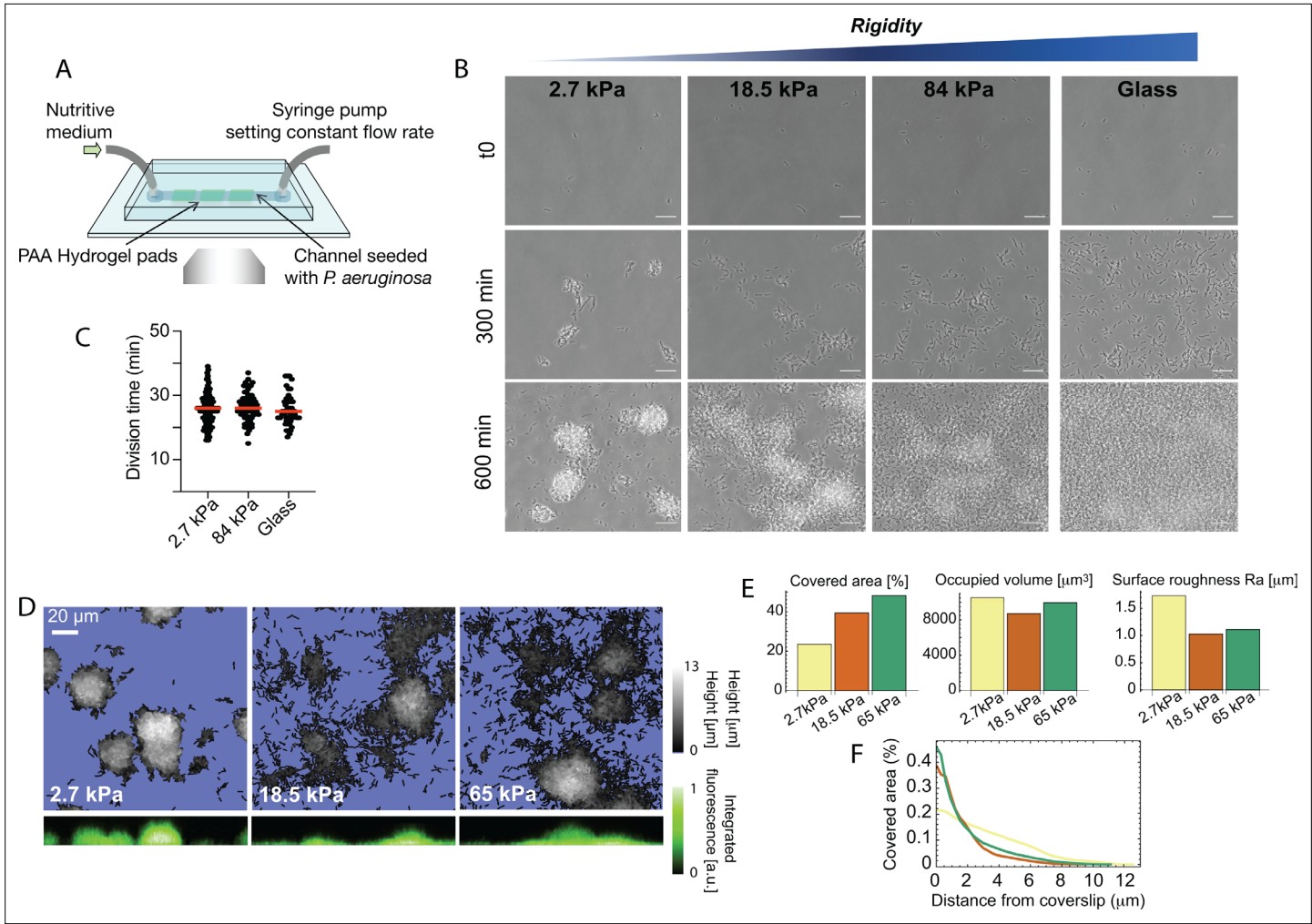

**Figure 1.** Bacterial microcolony formation depends on substrate rigidity. (**A**) Experimental setup: bacteria (*P. aeruginosa* strain PAO1) are imaged in a flow cell under constant flow of minimal medium. (**B**) After 10 hr, dense, isolated colonies form on soft PAA (2.7 kPa) while bacteria are more evenly distributed on stiff PAA (84 kPa), closer to what is observed on glass. Scale bars, 20 µm. (**C**) Bacterial growth is not impacted by substrate rigidity. (**D**) 3D reconstruction of colonies confirms their hemispherical shape on soft substrates. (**E**) Surface coverage is lower on soft substrates, but total volume of colonies is conserved, with a higher roughness value. (**F**) Fraction of area occupied by the bacteria as a function of the distance from the coverslip, showing flatter colonies on rigid substrates.

The online version of this article includes the following figure supplement(s) for figure 1:

**Figure supplement 1.** Mechanical characterization of hydrogels by AFM (**a**) comparison of elastic moduli measured by indentation and by microrheology.

**Figure supplement 2.** Morphology of microcolonies is strongly impacted by surface rigidity on PEG hydrogels.

**Figure supplement 3.** Substrate rigidity impacts early microcolony morphology in the absence of shear flow.

**Figure supplement 4.** In the T4P-deficient mutant PAO1 *pil A :: Tn*5, substrate rigidity does not significantly impact colony morphology.

## Substrate elasticity modifies bacterial colonization of PAA in a T4P-dependent manner

We first focus on the effect of substrate rigidity on early microcolony formation. Straightforward observations with phase-contrast imaging show a striking impact on the shape of microcolonies after a few hours (*Figure 1B* and *Video 1*): on the softest hydrogels (<10 kPa), bacteria form well-defined, dense hemispherical colonies; in contrast, on stiff hydrogels, bacteria are distributed in a thin layer covering most of the surface, a morphology closer to what we observe on glass. To rule out any effect due to changes in the bacterial growth rate, we quantified the division time of bacteria (*Figure 1C*), and the volume occupied by bacterial colonies after a few hours (*Figure 1D and E*) on different substrates:

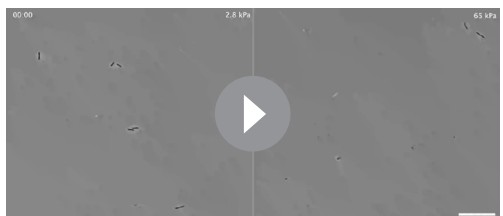

**Video 1.** surface colonization on 2.7 kPa (left) and 65 kPa (right) PAA hydrogels, imaged with phase contrast microscopy with one image/min over 6 hours. The two gels were included in the same microfluidic channel and imaged quasi-simultaneously. Scale bar, 20 m.
https://elifesciences.org/articles/81112/figures#video1

both were found to be unaffected, suggesting that bacteria develop and colonize substrates at the same rate irrespective of rigidity, but that the processes that drive their self-organization into colonies are modified. In contrast, a change in the morphology of the colonies could be confirmed by quantifying the characteristic roughness of the bacterial layer, which decreases as rigidity increases (*Figure 1E*), and the distribution of bacteria with the distance from the surface, which spreads further for soft hydrogels (*Figure 1F*). To control that this is a robust phenomenon driven by substrate elasticity rather than specific chemical interactions, we reproduced this assay using polyethylene glycol (PEG) hydrogels, which are chemically different from PAA but can span a similar range of rigidities. We obtained very similar results regarding the phenotype of colonies, which further confirms a role for the mechanical properties of the substrate in bacterial self-organization (*Figure 1—figure supplement 2*). Finally, because shear flow can orient polarly-attached bacteria, direct motility, disperse quorum-sensing molecules, and generally impact spatial organization into colonies, we carried out experiments on hydrogels immobilized at the bottom of wells, without any agitation of the above medium. Although long-term observations are rendered difficult in that case by swimming bacteria, we clearly observed the formation of denser colonies on softer PAA

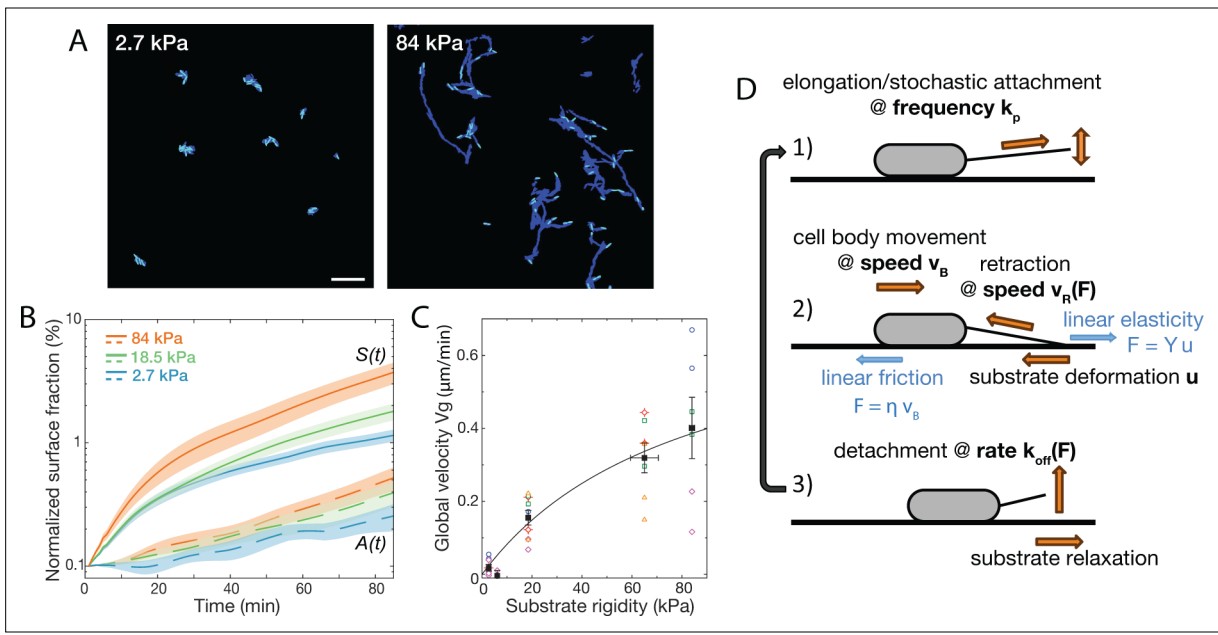

**Figure 2.** Bacterial surface motility is impaired on soft hydrogels. (**A**) Surface explored (dark blue) and current surface coverage (cyan) after 100 min on soft and stiff PAA surfaces. Scale bar: 20 μm. (**B**) Surface coverage A(t) (broken lines) and cumulative explored area S(t) (full lines) for all tested rigidities (the initial surface coverage $\langle t = 0 - 10 min \rangle$ was normalized to 0.1%). Shaded areas are standard errors of the mean (84 kPa: 6 data sets from independent experiments, 18.5 kPa: 10 data sets from five independent experiments, 2.7 kPa: 8 data sets from independent experiments). (**C**) Global bacterial motility $V_g$ averaged over the first 100 min, inferred from the difference between A(t) and S(t) (16 different surfaces, 6 independent experiments). The black line is the fit with the kinetic model using *equation A11*. with values $V_{max} = 0.77 \pm 0.35 \mu$ m.min$^{-1}$ and $E_0 = 84 \pm 68$ kPa. (**D**) Ingredients of the minimal 1D model for bacterial T4P-powered displacement.

The online version of this article includes the following source data and figure supplement(s) for figure 2:

**Source data 1.** Surface coverage and explored area vs time.

**Figure supplement 1.** Behavior of mutants $PAO1\ sadC :: Tn5$ and $PAO1\ wspR :: Tn5$ on soft (2.7 kPa) and stiff (84 kPa) PAA hydrogels.

(*Figure 1—figure supplement 3*), which further demonstrates that substrate stiffness modifies bacterial behavior after attachment in a wide variety of environments.

Since surface motility is known to be important for initial self-organization of PA, we hypothesize that it could play a role in the different microcolony shapes that we observe. This link was explored by carrying out experiments using a mutant deprived of type IV pili (T4P), and thus unable to twitch on surfaces (mutant PAO1 *pilA*::Tn5, *Figure 1—figure supplement 4*). In these assays, the dependence of microcolony morphology on substrate rigidity is abolished and bacteria form dense hemispherical colonies on all PAA substrates. We therefore conclude that T4P-mediated surface motility ('twitching') plays a key role in the rigidity modulation of microcolony formation by WT PAO1 on soft elastic substrates.

## Substrate elasticity modulates twitching motility

### Experimental results - global motility

To quantify the coupling between the elasticity of the substrate and the twitching motility of bacteria, we analyzed time-lapse phase contrast images of adhering bacteria in flow cells. These images allow segmentation of individual bacteria (SI subsection I.A) from the start of the acquisition (with a few isolated bacteria per field of view) until the transition to out-of-plane growth, after which individual bacteria cannot be easily separated anymore. From segmented binary images at early imaging stages (<100 min), we obtain the surface coverage $A(t)$ as the fraction of occupied pixels, and the cumulative explored area $S(t)$ as the fraction of pixels that has been explored at time $t$ (*Figure 2A*). The evolution of $A(t)$ reflects the exponential growth of initially attached bacteria on the surface, as well as potential attachment and detachment events during the acquisition. However, in our experiments initial surface coverage is extremely low, and at early times the number of bacteria in the clean flowing medium is negligible so that attachment events are rarely observed. We can thus consider that

$$\frac{dA}{dt} = (k_{di} - k_{de})A(t). \tag{1}$$

The bacterial division rate $k_{di}$ does not depend on the substrate (*Figure 1C*), and was measured for each experiment ($k_{di}^{-1}$=27.8 ± 1.4 min). *Figure 2B* shows the experimental time evolution of $A(t)$ depending on the gel rigidity, which can indeed be well described by a simple exponential (straight line in the semi-log presentation with slope $k_{di} - k_{de}$). The slope of $A(t)$ slightly increases with gel rigidity, suggesting a higher detachment rate $k_{de}$ on softer hydrogels at early acquisition times.

Compared to $A(t)$, there is a quantitatively much larger dependence of the cumulative surface coverage $S(t)$ on the substrate rigidity (*Figure 2B*). We propose that this result directly reflects a change of the global motility $V_g$ of bacteria on the surface. Indeed, neglecting bacterial surface attachment, the evolution of $S$ can be written as

$$\frac{dS}{dt} = k_{di}A + V_g w_b N = (k_{di} + \frac{V_g}{l_b})A. \tag{2}$$

where $N$ denotes the number of bacteria on the surface, and the typical size of a rod-shaped bacterium is $w_b l_b$ (width × length), so that the occupied area is $A = N w_b l_b$. Here we have assumed that bacteria tend to move along their major axis, neglecting reorientations, based on previous findings about the polar localization of T4P (*Cowles and Gitai, 2010*; *Jin et al., 2011*; *Kühn et al., 2021*) and our own observations. (Considering that bacteria can move in any direction would lower velocity values by a factor $\sqrt{\frac{l_b}{w_b}} \approx \sqrt{3}$, and not change significantly the coming discussion.) The average bacterial size was measured in each experiment ($l_b$ =2.8 ± 0.13 $\mu m$, $w_b$ =0.8 ± 0.13 $\mu m$). For each monitored position, we determined $dS/dt$ and $A(t)$ experimentally. The global bacterial velocity $V_g$ was then estimated using *Equation 2*, by averaging over the first 100 experimental time points.

During the first 100 min under flow, $V_g$ exhibits a clear dependence on substrate elasticity (*Figure 2C*). Motility values are close to zero on very soft substrates (3–6 kPa), and progressively increase to reach $\sim 0.5 \pm 0.25$ μm/min on the stiffest hydrogels tested in this study (84 kPa). We wondered whether this dependence of surface motility on substrate elasticity could result from intracellular modifications in response to bacterial surface-sensing. In *P. aeruginosa*, two main surface-sensing systems have been unveiled so far: the Pil-Chp system that involves T4P retraction-mediated force sensing *Webster et al., 2021*, and the Wsp system believed to be activated by cell envelop stress *O'Neal et al.*,

*2022*; both systems have been shown to activate biofilm formation pathways following bacterial adhesion *Chang, 2017*. Sessility and matrix production are promoted by increasing intracellular c-di-GMP levels, which production is catalyzed by diguanylate cyclases (DGCs). We carried out experiments with wspR and sadC mutants, two DGCs known to be involved in the surface-sensing response. Although impaired in c-di-GMP regulation, both mutants still exhibited a stiffness-dependent twitching motility (*Figure 2—figure supplement 1*). Even though other DGCs might be involved, these initial results suggests that our observations at early timescales could be mainly governed by the mechanical interaction of bacteria with their substrate, with gene regulation playing a secondary role. This rational motivates the simple kinetic modeling presented next.

## Minimal kinetic modeling

Because a difference in twitching velocity is observed almost immediately upon attachment of bacteria onto the surface, a simple hypothesis could be that a modulation of the twitching efficiency arises through mechanical factors – the interplay between the T4P extension/retraction mechanism and the linear elasticity of the substrate – without the need for mechanotransduction mechanisms. To test this minimal hypothesis, we have developed a kinetic model, schematically described on *Figure 2D*.

Briefly (more details can be found in appendix 1), we consider a bacterium adhering onto an elastic substrate with a single effective pilus. The pilus is modelled as a rigid inextensible filament (*Beaussart et al., 2014*) and attaches to the substrate via its extremity with a typical adhesion size $\lambda$. This simple choice is motivated by microscopic observations of pilus straightening over its whole length during retraction *Talà et al., 2019* and an estimation of the pilus attachment spot size of $\approx 1$ nm from traction force microscopy measurements *Koch et al., 2022*. Note, that multiple attachments of pili over extended regions to the substrate have also been suggested *Lu et al., 2015*. However, in our simple approach, we do not consider this possibility. The cell actively retracts its pilus until it detaches from the substrate with the force dependent velocity $v_R(F) = v^0(1 - \frac{F}{F_R})$(*Marathe et al., 2014*), where $F$ denotes the tensile load on the pilus, $F_R$ the retraction stall force and $v^0$ the retraction speed at zero load. Assuming linear elasticity, the tensile load $F$ is related to the substrate displacement $u$ at the pilus adhesion patch by $F = Yu$, where $Y \sim E\lambda$ and $E$ is the Young's modulus of the substrate. Since the typical size of the bacterial body $l_b$ is much larger than $\lambda$, we neglect the deformation of the substrate induced by the bacterial body. Instead we assume that the pilus tension leads to a forward sliding of the bacterial body with a linear force-velocity relationship $v_B(F) = v^0 \frac{F}{F_B}$ (see SI subsection II.B and *Sens, 2013*), reducing the substrate deformation and the load in the pilus. Here, the ratio $\eta = \frac{F_B}{v_0}$ denotes the mobility constant of the cell on the substrate. With this model, the evolution of the pilus tension $F$ is thus given by

$$\frac{dF}{dt} = Y\frac{du}{dt} = v_R(F) - v_B(F) \tag{3}$$

which is solved by

$$F(t) = F_0 \left(1 - e^{-\frac{Y v^0}{F_0}t}\right) \tag{4}$$

with the naturally arising force scale

$$F_0 = \frac{F_B F_R}{F_R + F_B}. \tag{5}$$

From $\frac{dx_B}{dt} = v_B(F)$ we obtain the bacterial sliding distance during the pilus retraction

$$x_B(t) = \frac{F_0}{F_B}\left[v^0 t + \frac{F_0}{Y}\left(e^{-\frac{Y v^0}{F_0}t} - 1\right)\right]. \tag{6}$$

While retracting the pilus will detach with a rate constant $k_{off}$ from the substrate. Assuming a force-independent off-rate constant $k_{off} = k_{off}^0$ (and hence a mean pilus adhesion time $\left(k_{off}^0\right)^{-1}$) and a pilus retraction frequency $k_p$, we obtain an effective bacteria velocity:

$$v_{eff} = k_p \langle x_B \rangle = k_p k_{off}^0 \int_0^\infty x_B(t)\, e^{-k_{off}^0 t}\, \mathrm{d}t = V_{max} \frac{E}{E + E_0}\,. \tag{7}$$

Here, $\langle x_B \rangle$ denotes the mean bacterial sliding distance per pilus retraction event and $V_{max}$ denotes the maximum effective speed a bacterium can reach on a given substrate at infinite rigidity. It is given by

$$V_{max} = v^0 \frac{k_p}{k_{off}^0} \frac{F_R}{F_B + F_R}\,. \tag{8}$$

$E_0$ denotes the rigidity at half-maximal speed and is given by

$$E_0 = \frac{F_B F_R k_{off}^0}{(F_B + F_R) v^0 \lambda}\,. \tag{9}$$

Fitting (A11) against experimentally measured effective bacterial velocities $V_g$ provides a quantitative description of the data for $V_{max} = 0.77 \pm 0.35$ μm.min⁻¹ and $E_0 = 84 \pm 68$ kPa. The error estimates for $V_{max}$ and $E_0$ were calculated directly from the co-variance matrix of the fit function and the variance of residuals (chi-squared sum divided by the number of degrees of freedom) and are reflective of the wide scattering of measured velocities between different experiments. Conversely, we can estimate $V_{max}$ and $E_0$ from values of the parameters used in the model: assuming a typical pilus retraction speed $v^0 = 1$ m.s⁻¹ (*Marathe et al., 2014*), a stall force of the order $F_R = 100$ pN (*Marathe et al., 2014*; *Koch et al., 2022*), a pilus off-rate constant $k_{off}^0 = 1$ s⁻¹ (*Talà et al., 2019*), a contact size of $\lambda = 1$ nm (*Koch et al., 2021*), a high friction surface with $F_B = 1$ nN and a typical pilus retraction frequency (Here we assume that one single effective pilus is active during a retraction event. Using a typical pilus length of 5 μm with retraction speed of $v_0 = 1$ μm.s⁻¹ we obtain a duration of 5 s per retraction and a retraction frequency of 0.2 s⁻¹) of $k_p = 0.2$ s⁻¹ we obtain $V_{max} \sim 1$ μm.min⁻¹ and a substrate rigidity at half maximum speed of $E_0 = 100$ kPa, which are within 30% of the fitted values.

In addition, our model [*Equation 8*] shows that two separate effects translate a (fast) load-free microscopic pilus retraction speed $v^0$ into a (slow) macroscopic bacterial speed $V_{max}$. First, the bacterium only translocate during a fraction of the pilus cycle of extension and retraction $\frac{k_p}{k_{off}}$ over the substrate. Second, the pilus retraction speed slows down in a load dependent manner $F_R/(F_B + F_R)$. Both effects together reduce the local speed by an order of magnitude from $\mu m.s^{-1}$ to $\mu m.min^{-1}$.

Together this demonstrates that our experimental results on bacterial effective motility on elastic substrates can be interpreted as the result of a simple interplay between the pilus retraction mechanism, the deformation of the elastic substrate, and the friction of the bacterial body on this substrate.

## Analysis of individual trajectories

The simple approach presented *Figure 2* yields a population-averaged value of the bacterial velocity $V_g$. Yet, bacterial populations can be heterogeneous, and moreover the model we have used *Equation 2* to determine $V_g$ relies on a number of strong assumptions, such as neglecting detachment and re-attachment events. To go further in dissecting bacterial motility on PAA substrates, we developed a segmentation and tracking protocol in order to obtain the individual tracks of all the bacterial cells visible over the course of the acquisiton (*Figure 3A*, *Video 2* and Materials and methods for details). This thorough approach allows us to measure the velocity associated with each 1 min displacement step.

Can a characteristic twitching velocity be defined for all bacteria? or do phenotypically distinct populations of slow and fast bacteria cohabit on the surface? To answer these questions, we labeled each track, defined as the displacement of a bacterium between two successive division events (*Figure 3A*). We expected track duration to be similar to the characteristic division time shown on *Figure 1C*. However, we obtained a bimodal distribution, with two peaks centered at times unaffected by the substrate rigidity: one peak indeed centered on the division time (~27 min), and a second one that corresponds to bacteria spending 5–10 min on the surface before detaching (*Figure 3—figure*

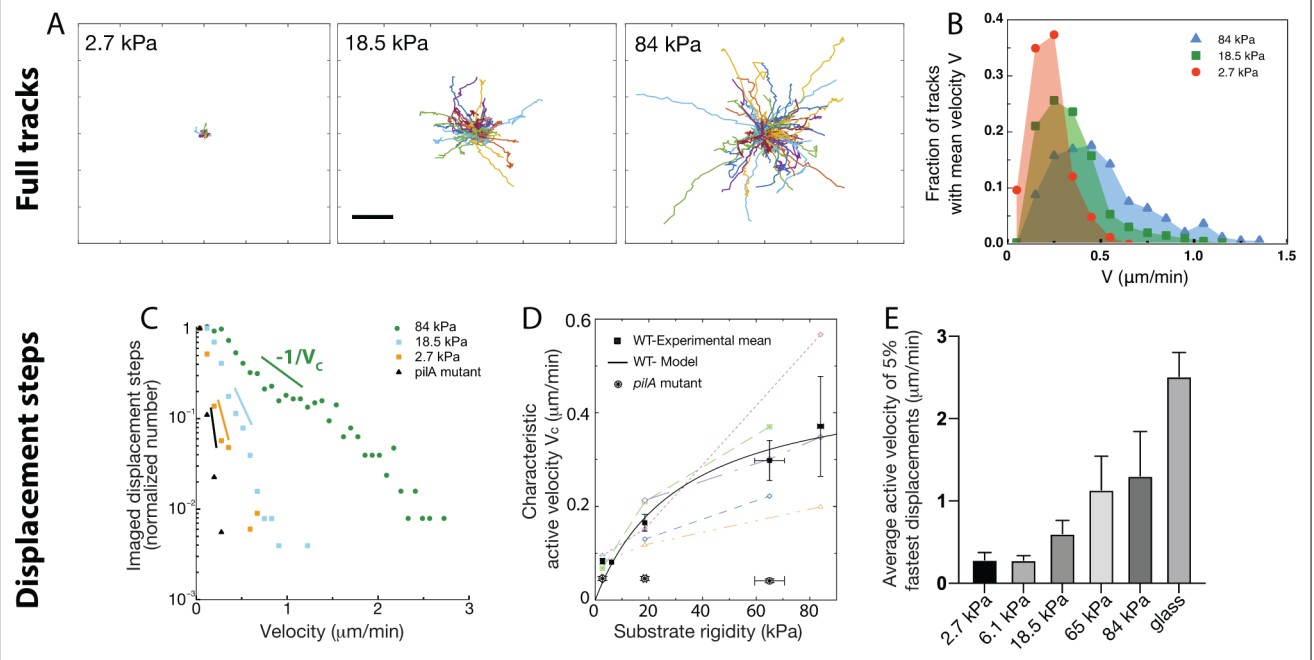

**Figure 3.** Twitching motility depends on substrate rigidity and is highly distributed in the bacterial population. Analysis of full tracks (top row) and 1 min displacement steps (bottom row). (**A**) Individual bacterial tracks on soft (2.7 kPa), intermediate (18.5 kPa) and stiff (84 kPa) PAA during the first 3 hours after bacterial inoculation (total number of tracks is respectively 60, 123, and 175). Scale bar: 10 μm (**B**) Mean track velocity distribution for different values of the substrate rigidity. Only full tracks were considered (corresponding to the right peak in *Figure 3—figure supplement 1*). 84 kPa: 330 tracks from 2 independent experiments, 18.5 kPa: 394 tracks from 3 independent experiments, 2.7 kPa: 83 tracks from 2 independent experiments. (**C**) Normalized velocity distributions for the whole bacterial population on different PAA surfaces. The exponential decrease yields a characteristic active velocity $V_C$ on each substrate. Displacement steps were measured every minute for 100 min, and two positions were acquired on each rigidity. The average of T4P-defective mutant on all surfaces is shown as a reference. (**D**) Characteristic active velocity values $V_C$ obtained by fitting velocity distributions (6 independent experiments, 16 different surfaces). Values measured on different surfaces in a single experiment (same channel) are shown with the same symbols and connected. Black squares(circled stars) are mean values obtained for the WT(*pilA* mutant), and error bars show the SEM. The black line is the fit of WT values with the kinetic model-derived **equation A11**, with $V_{\max} = 0.48 \pm 0.12$ μm.min⁻¹ and $E_0 = 32 \pm 30$ kPa. (**E**) Average velocity values of the top 5% fastest displacement steps for different substrates. Error bars are standard errors.

The online version of this article includes the following source data and figure supplement(s) for figure 3:

**Source data 1.** Instant velocity values from bacterial tracking.

**Figure supplement 1.** Tracking result reveals bacterial sub-populations on hydrogel substrates.

**Figure supplement 2.** Anaysis of the mean track velocity.

**Figure supplement 3.** Validation of the fitting of displacement steps distributions with single exponentials.

**Figure supplement 4.** Statistical analysis of $V_g$ (**A**) and $V_c$ (**B**) for various gel rigidities.

**Figure supplement 5.** Characteristic twitching velocity measured on PEG hydrogels is similar to the one measured on PAA hydrogels under identical experimental conditions.

*supplement 1A*). The velocity distribution corresponding to each population is similar (*Figure 3—figure supplement 1B*). This observation is consistent with a phenotypical difference between daughter cells, in agreement with the results of *Laventie et al., 2019* Indeed, the duration of the track before detachment tends to be shorter on soft substrates, but the fraction of bacteria that detach from the substrate ($35 \pm 2$ %) is independent of the substrate rigidity. We thus assume that at early experimental timepoints, after moving in sync with the first daughter cell, the second one sometimes detaches from the substrate (about 70% of the time). This feature might change at later timepoints - bacterial tracking was interrupted at the onset of 3D spatial organization (~100 min on the softest hydrogels, see next section for a full description).

Considering only adhering offsprings, for which full tracks were recorded, we normalized tracks with respect to their initial position, yielding *Figure 3A*. These homogeneous radial distributions confirm that shear does not influence bacterial orientation in our experiments. As expected, track

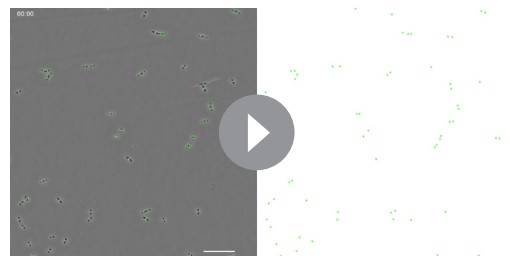

**Video 2.** Principle of the image processing for the tracking of individual bacteria. Glass surface colonization under controlled shear flow followed over 90 min (1 frame/minute) by phase contrast microscopy. Left, the registered phase contrast image is superimposed with the center of mass of the detected cells after segmentation (green dots). Right, tracks of the detected cells (obtained with the simple LAP tracker of the Trackmate ImageJ plugin), color-coded as a function of the track final length. Scale bar, 20 m. https://elifesciences.org/articles/81112/figures#video2

extension becomes larger as substrate rigidity increases. The distribution of the mean velocity of tracks does not allow us to distinguish different bacterial sub-populations: it reaches higher values on stiffer hydrogels, but it is broad, continuous with an exponential decay (reflecting the diversity of behaviors expected in a population of cells; *Figure 3B*). In addition, for each track the standard deviation of this mean velocity is comparable and proportional to its mean (*Figure 3—figure supplement 2*), suggesting a stochastic distribution of twitching steps within a given trajectory.

Focusing next on 1 min displacement steps, and pooling all monitored events during the first 100 min of experiments (which provides more data than only analysing full tracks within the same time window), we obtained the typical velocity distributions shown on *Figure 3C*. These distributions further confirm that bacterial displacements are very heterogeneous, and present an exponentially decreasing tail which fitting yields a characteristic velocity $V_C$ for the bacterial population on a given substrate, that is:

$$N(V > V_0) = N_0 \exp\left(-\frac{V - V_0}{V_C}\right), \tag{10}$$

($V_0$=0.08 µm/min denotes a visual cutoff for low velocities).

To rationalize the meaning of $V_C$, we reasoned that displacement steps are the sum of a passive velocity due to bacterial elongation, local reorganizations and experimental noise, and an active velocity powered by T4P. The velocity distribution obtained using a *pilA* mutant is purely exponential, and was used to determine the characteristic passive motility, which does not significantly depend on substrate rigidity ($V_C(pilA) = V_{C,p} = 0.044$ µm/min, see *Figure 3D*). In the case of motile strains, assuming that active and passive displacements are incoherent, our numerical calculations (*Figure 3—figure supplement 3*) show that in the limit $V_C > V_{C,p}$, the fitted characteristic velocity $V_C$ obtained as described above reflects the active twitching motility of the population, and is not significantly affected by passive movements. A detailed justification for our analysis of the probability distributions of displacement steps is given in the Methods section.

This analysis, which does not rely on any strong assumption, yields active velocity values for the WT strain (*Figure 3D*) in very good qualitative agreement with the global velocity analysis described earlier ($V_g$, *Figure 2C*, and *Figure 3—figure supplement 4*). Again, our kinetic model describes the data quantitatively with values very close to the ones fitted and calculated in the previous subsection ($V_{\max} = 0.48 \pm 0.12$ µm.min$^{-1}$ and $E_0 = 32 \pm 30$ kPa). The large error bars on fitting parameters reflect the dispersion of experimental measurements, despite our efforts to reproduce identical experiments. However, velocity values measured in a given experiment always exhibit a similar dependence on substrate rigidity, that is a clear increase of motility as rigidity increases. In addition, we characterized the velocity of the 5% fastest bacterial displacement steps (*Figure 3E*) on each type of substrate. This analysis confirms the dependence of twitching velocity on substrate rigidity, but also yields higher velocity values, in good quantitative agreement with those reported in the literature using other experimental approaches (*Talà et al., 2019*), which might be biased toward more active bacteria. Finally, this approach was used to quantify bacterial motility in experiments on PEG hydrogels (shown *Figure 1—figure supplement 1–pp.*), which confirms the very similar bacterial behavior on the two kinds of substrates we have used (*Figure 3—figure supplement 5*).

# Rigidity-modulated bacterial motility governs the spatial characteristics of early surface colonization

## In-plane to 3D transition of emerging colonies

To understand the way rigidity-modulated bacterial motility impacts the process of microcolony formation, we studied in details the way colonies transition to out-of-plane growth. Several experimental and theoretical approaches have been developed in the past to decipher this process: for confined colonies, the switch from planar to 3D growth takes place when it becomes energetically too costly to push neighboring cells outwards. In that case, the adhesion forces between the bacteria and their underlying substrate play a key role: strongly adhering bacteria transition to 3D colonies earlier in their development (*Duvernoy et al., 2018*; *Grant et al., 2014*). In our experiments, there is no strong vertical or lateral confinement: bacteria can move on the substrate or away from it, so that cells stemming from a given progenitor do not necessarily stay in contact with each other. However, the twitching velocity determines how much cells, on average, move away from one another between two successive division events, thereby creating space to accommodate new offsprings on the surface.

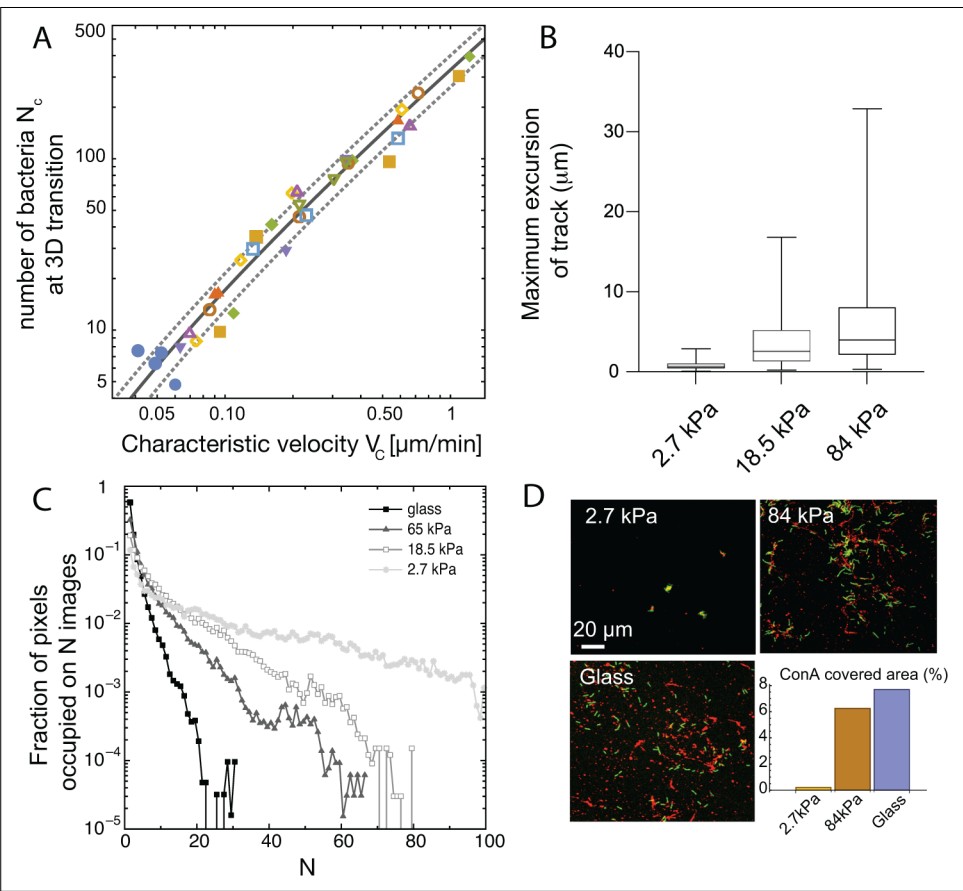

**Figure 4.** Spatial structuring of surface colonization is impacted by substrate rigidity through twitching velocity. (**A**) Size of microcolony (in number of bacteria $N_c$) at the 2D to 3D transition as a function of the center-of-mass characteristic velocity $V_C$ defined in *Figure 3A*. Markers are experimental data from 9 different experiments, each with different substrates including glass (hence leading to higher values that in *Figure 3B*). Blue dots are data obtained with the pili-deficient mutant pilA:: Tn5. Lines, kinetic model for $\langle \gamma \rangle$ (solid line) ± its standard deviation (dotted lines). (**B**) Distribution of track lengths of full trajectories as a function of the substrate rigidity. (**C**) Distribution of occupation occurrence on each image pixel as a function of rigidity, showing a much more heterogeneous occupancy on soft substrates. (**D**) ConA staining (red) of EPS deposition during cell (green) exploration of the surface.

The online version of this article includes the following figure supplement(s) for figure 4:

**Figure supplement 1.** EPS staining with concanavalin A highlights matrix deposits on the surface of substrates of different stiffnesses after 8h of surface colonization.

To investigate the possible link between twitching motility and 2D to 3D transition of growing microcolonies, we sought to determine $N_c$, the number of adhered cells in a progeny (i.e. stemming from successive divisions of a given bacterium) when the 2D to 3D transition takes place, as a function of the twitching velocity. For softer substrates, all bacteria can be imaged, and $N_c$ is directly measured; we also determined the average number of colonies per unit area. On stiffer substrates, it is impossible to track all bacteria stemming from a mother cell, since they are very motile and sometimes move out of the field of view. We assume that bacteria from other progenies are equally likely to move inside the field of view, so that measuring the number of bacteria on the image at $t_c$ (the time at which transition to 3D is first observed), divided by the average microcolony density determined on softer substrates earlier gives a good approximation of $N_c$. *Figure 4A* shows $N_c$ as a function of the center-of-mass characteristic velocity $V_C$ determined above (*Figure 3D*), on different substrates and for nine different experiments. $N_c$ consistently increases with the twitching velocity, indicating a strong correlation between the twitching efficiency and the shape of early colonies and shedding light on our initial observations of variations in microcolony morphology as a function of the substrate rigidity (*Figure 1B and D–F*).

To decipher the link between $N_c$ and $V_C$, we have built a simple kinetic model with a single unknown parameter (see appendix 2 for details). Briefly, we assume that the 2D to 3D transition takes place when the area occupied by bacteria reaches a fraction of the equivalent 'microcolony size', defined as the characteristic area explored by bacteria in a progeny. Assuming that bacteria explore the surface through a random walk with persistence (*Marathe et al., 2014*), the characteristic area accessible to bacteria in a microcolony over time can be written as $a(t) = a_0(1 + \alpha V_C t)$ where $a_0$ is the area of one bacterium and $\alpha$ is a parameter related to the properties of the random walk. Our experimental data show that not only the velocity, but also the contour length of the trajectories of bacteria increases with the rigidity since the duration of these trajectories is mostly constant (*Figure 4B* and *Figure 3— figure supplement 1*). Area $a(t)$ is related, but not equal, to the area over which the microcolony spreads. Indeed, bacteria are not evenly distributed within the microcolony area, and we observe strong local density fluctuations. If we now consider an exponential growth of the number of bacteria on the surface due to the balance of cell division and detachment, it follows that the increase in the number of cells, and hence the area required to accommodate these cells on the surface, grows faster than the accessible area, driving a transition to 3D growth. Expressing the number of cells $N_c$ in the microcolony at the time when this transition stochastically occurs leads to the following dependence as a function of $V_C$:

$$N_c = 1 + \gamma V_C \log(N_c) \tag{11}$$

$\gamma$ is an unknown parameter related to the properties of the random walk and the growth rate of bacteria on the surface that can be measured for each experiment. On *Figure 4A*, we have plotted the corresponding curve using the average of experimental values for $\gamma$ (solid line) ± their standard deviation (dotted lines). We observe an excellent agreement between this simple kinetic model and our experimental data over a wide range of velocities, including the T4P deficient mutant and the WT strain adhering on glass. This hints that elasticity is a key factor shaping the organization of early colonies on elastic substrates, and that it is the main determinant of the colony shapes observed in our experiments on chemically identical substrates with varying rigidities (rather than energy minimization whereby bacteria would either favor adhesion to other cells or to the substrate depending on its rigidity).

## Surface decoration by extracellular matrix

One consequence of the modulation of twitching efficiency by the substrate elasticity could be a variation in matrix distribution on the surface. Indeed, *P. aeruginosa* can secrete an extracellular matrix mostly composed of exopolysaccharides (EPS), which was shown to result in the deposition of 'trails' by twitching bacteria on glass substrates. By mediating the attachment of the cell body to the underlying substrate such deposits are inferred to facilitate further colonization by bacteria and to impact microcolony formation (*Liu et al., 2007*; *Zhao et al., 2013*). To investigate matrix deposition on hydrogel substrates, we introduced a fluorescent dye (lectin concanavalin A, see Materials and methods) in the nutrient medium infused in our device. The main component of PAO1 matrix, psl (*Jackson et al., 2004*), is rich in mannose, that conA specifically binds (*Ma et al., 2007*).

For high stiffness substrates where bacteria explore the surface efficiently, this staining confirms that trails of matrix decorate a significant fraction of the surface; on the contrary, nearly immobile bacteria on soft substrates accumulate matrix locally, leaving most of the surface unmodified (*Figure 4C and D*). This difference in matrix distribution is maintained at a later stage of surface colonization (*Figure 4—figure supplement 1*). While on rigid hydrogels, most of the surface is covered by bacteria-secreted matrix, lectin staining on soft hydrogels is only present on compact colonies separated by regions completely devoid of EPS. While proper quantification of the total amount of matrix produced in each case is difficult (staining efficiency might be impacted by lectin diffusion inside dense colonies), our results confirm that substrate rigidity impacts bacterial propensity to modify hydrogel substrates via matrix deposits.

## Substrate rigidity affects bacterial mixing

Real-life biofilms generally comprise several species: pathogens can compete or help each other (*DeLeon et al., 2014*; *Orazi and O'Toole, 2017*), and commensal strains protect organisms from detrimental ones (*Aoudia et al., 2016*). To further investigate how modulation of surface colonization with rigidity impacts the structure of forming biofilms, we studied the model co-colonization of hydrogels by two PAO1 strains constitutively expressing two different fluorescent proteins. Beside their fluorescence, the two strains exhibit identical properties (motility, division rate, etc.), similar to that of WT PAO1. Through fluorescent confocal imaging, the strains were spectrally separated to study their spatial distribution at different stages of surface colonization. As expected, rigidity-modulated motility

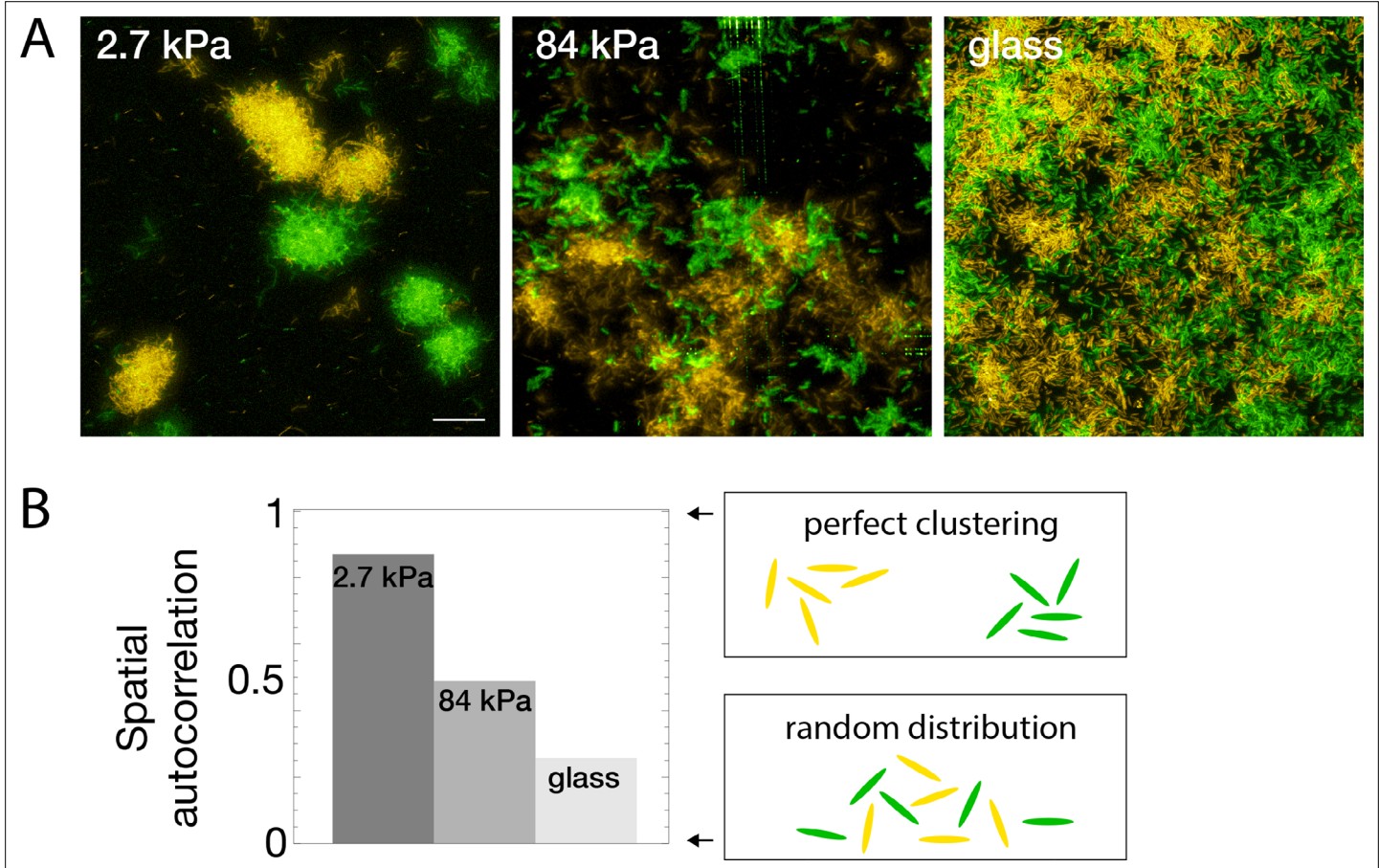

**Figure 5.** Bacterial spatial distribution is impacted by substrate rigidity. (**A**) Images of surfaces seeded with a 1:1 mixture of constitutively fluorescent bacteria expressing GFP or YFP show mostly monoclonal colonies on soft hydrogels, and mixed bacteria on rigid substrates (3D-rendering obtained by stacking images). Scale bar, 20 μm.(**B**) Spatial correlations quantified via Moran's I index.

The online version of this article includes the following figure supplement(s) for figure 5:

**Figure supplement 1.** Bacterial spatial distribution as a function of substrate rigidity.

impacts the co-colonization of the hydrogels from early stages (*Figure 5—figure supplement 1*): on rigid substrates, high motility promotes mixing of the offsprings of different cells, resulting in a spatial distribution of the two strains close to random (a residual correlation between the colour of neighboring cells is always found due to the presence of cells that have just divided). Conversely, nearly immobile cells on soft substrates exhibit strong correlations between neighboring cells, which mostly arise from a single progenitor cell. This striking difference in strain mixing during surface co-colonization is maintained at later stages of biofilm formation: on soft substrates, quasi-monoclonal colonies with complete spatial segregation are observed, while biofilms forming on rigid surfaces exhibit a close-to-random distribution of the two strains at the 10-μm scale (*Figure 5A*). To quantify this effect, we have used Moran's I index, a statistical tool designed to quantify the spatial clustering of species. It provides a measure of the local spatial correlations and takes values ranging from 1 (perfectly correlated values) to –1 (perfectly anti-correlated values), with 0 corresponding to a spatially random distribution of the variable (see Materials and methods for details). The resulting quantitative analysis (*Figure 5B*) confirms the decisive impact of rigidity on the structure of mixed biofilms, with potentially far-reaching consequences on the interactions of different strains in multi-species biofilms.

## Surface rigidity impacts gene expression

Cell-cell communication, either via exported molecules or by direct contact is crucial during biofilm development (*Shrout et al., 2011*). Modifications of bacterial distribution as described above could thus likely impact gene regulation in surface-attached bacteria. To start addressing this complex question, we focused on the expression level of cyclic-di-GMP (c-di-GMP), a second messenger that controls the motile-to-sessile transition in *P. aeruginosa* (*Rodesney et al., 2017*). We used a post-transcriptional fluorescent reporter build on the promoter of the gene *cdrA*, which encodes an exported protein involved in matrix cohesion, upregulated during biofilm formation by PAO1 (*Reichhardt et al., 2018*). The PcdrA-gfp intracellular reporter provides a measure of the integrated production of CdrA with a ~40-min delay between expression of the gene and fluorescence detection (*Rybtke et al., 2012*). *Figure 6* shows how *crdA* expression is modulated by rigidity on 4 substrates included in the same microfluidic device. For this reporter, the degradation rate of gfp occurs over several hours, and its dilution due to growth and division of bacteria occurs at the same rate on all surfaces (see *Figure 1C*). The increase rate of the fluorescence signal is thus a direct proxy to the expression rate of gene cdrA, and thus to the changes in c-di-GMP level.

During a first phase of surface colonization, fluorescence remains low on all surfaces. The signal subsequently starts increasing linearly, roughly at the same time for all surfaces (within the uncertainty of fluorescence quantification, that is ≈10 min). This second phase ends with the onset of a plateau, again around the same time for all surfaces, at the end of the exponential growth of bacteria adhered on the surface, possibly as a result of oxygen depletion in the flowing medium that would be sensed simultaneously on all surfaces (*Figure 6—figure supplement 1*). This linear increase in fluorescence directly translates into a constant production rate of CdrA that can be compared for the 4 surfaces (*Figure 6C*): our analysis shows a marked increase in CdrA expression with the substrate rigidity.

## Discussion

In this study, we have designed an experimental approach to investigate early microcolony formation by *P. aeruginosa* on hydrogels with different elastic moduli, under constant flow rate. By continuously imaging surface-attached bacteria in situ, we show that substrate rigidity influences the twitching motility of individual bacteria, therefore strongly impacting the process of microcolony formation. Through two different analyses of the surface motility of the bacterial population, either via the global evolution of the explored area or via the tracking of individual cells, we find that the characteristic twitching velocity increases with substrate stiffness (from 0.02 to 0.4 μm/min when rigidity goes from ~3 to 80 kPa).

The encounter between bacteria and a substrate generates mechanical stress. Deciphering surface-sensing, that is understanding how the mechanical feedback resulting from this interaction translates into chemical signals that will in turn tune bacterial behavior has been the focus of a lot of recent research. It is now clear, for instance, that T4P contraction acts as a force sensor that transmits signals to the bacterium at the single-cell level (*Webster et al., 2021*), and triggers a response that involves

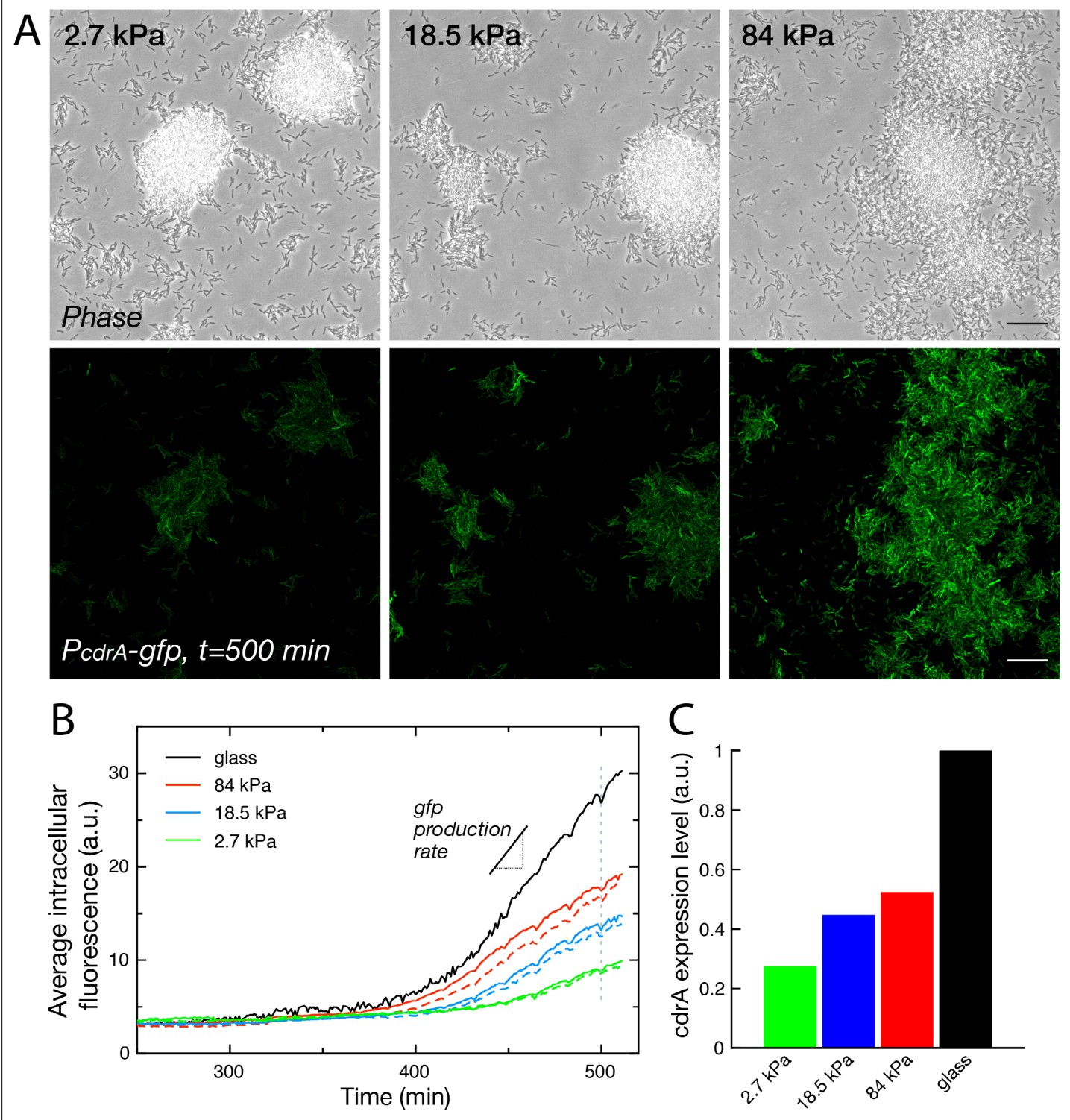

**Figure 6.** Substrate rigidity influences gene expression levels (**A**) Colonies grown from a modified PAO1 strain bearing the plasmidic $P_{cdrA} - gfp$ reporter imaged in phase contrast (top) and fluorescence (bottom), after 500 min under flow. Scale bars, 20 µm. (**B**) Average intracellular gfp fluorescence as a function of time, on different PAA surfaces and on glass. Broken lines are duplicate positions on a given surface. (**C**) Average cdrA expression level, obtained from a linear fit of (**B**).

The online version of this article includes the following figure supplement(s) for figure 6:

**Figure supplement 1.** Fluorescence intensity from a PAO1 strain expressing a $P_{cdrA} - gfp$ plasmid over time, showing a plateau of fluorescence expression after ≈500 min.

an increase in c-di-GMP level (*Armbruster et al., 2019*). Furthermore, recent results suggest that pili can differentiate substrate rigidity, yielding a maximum response for stiffness values ~300 kPa *Koch et al., 2022*.

However, in our experiments a difference in bacterial motility can be observed almost immediately upon surface adhesion to soft or rigid PAA, and this behavior is not modified in mutants impaired in c-di-GMP regulation (*wspR* or *sadC*). We thus propose a physical model to account for the modulation of the twitching motility. This 1D model is based on a force balance between (i) a pilus that extends, attaches and retracts with a defined frequency; (ii) the deformation of the underlying substrate at the pilus tip upon retraction; and (iii) the friction force due to adhesion of the bacterial body when it is dragged across the surface at the other end of the pilus. In this balance, the detachment rate of the pilus tip from the substrate is a key parameter in the resulting bacterial velocity. We have assumed a force-independent off-rate constant for the pilus. In a more complex scenario, the contact between the pilus and substrate may act as a slip bond or a catch bond. For completeness we show some numerical results for slip and catch bond behavior in the SI (section I.D), which do not increase however the quality of fit between experimental and theoretical velocity data. In addition, although we have explored the possibility that substrate rigidity, which is directly correlated to the mesh size of the hydrogel network, could impact the frequency of attachment of T4P, this was not necessary to efficiently account for the variation in motility we observe, which we instead solely attribute to the elastic deformation of the substrate.

Strikingly, our minimal mechanistic model thus suggests that a variety of observed phenomena (3D structure of colonies; EPS deposition on the surface; strain mixing during co-colonization) can all derive from a modulation of the efficiency of pili activity by the deformability of soft substrates. This purely mechanical model may be of particular importance for surface colonization, since the adaptation of bacterial behavior to the environment can thus be instantaneous - possibly a key to PA ability to efficiently colonize extremely different microenvironments.

While this model is sufficient to account for our observations (twitching velocity, microcolony formation), it certainly does not rule out a regulatory response of the bacteria, which probably takes place in parallel. Such a response can happen on two levels: at the single-cell level, mechanotransduction processes mediated for instance by adhesion and retraction of T4P can influence gene expression at short timescales (~1 hour) *Armbruster et al., 2019*; *Song et al., 2018*. At longer timescales, in developing microcolonies, cell-cell interactions could in turn modulate the bacterial transcriptome, which depends on microcolony characteristics (e.g. shape, cell density, matrix content) *Livingston et al., 2022*. Our attempt at quantifying c-di-GMP expression using a fluorescent intracellular reporter does evidence a difference in bacterial regulatory response depending on substrate stiffness. While the level of expression of the gene is clearly impacted by the substrate rigidity, differences in expression level are detected only 6–7 hr after the onset of surface colonization, with a first phase characterized by low c-di-GMP level on all surfaces. This timeframe suggests that the difference in gene expression that we observe is probably not due to a direct sensing of the substrate rigidity by individual bacteria, but rather a consequence of their organization into more or less dense colonies. Of note, when the increase in c-di-GMP takes place bacteria have stopped twitching and immobilized into colonies. We do not observe the early increase in c-di-GMP described in the litterature, possibly because we initially only track a very small number of bacteria on the surface, and the expression signal is stochastic. Further investigating c-di-GMP expression in WT and mutant strains upon adhesion to mechanically different substrates could help reveal which pathways are differently activated on soft substrates.

Interestingly, our results show that microcolony phenotype may not be indicative of a specific c-di-GMP regulation. The dense colonies observed on soft hydrogels correspond to lower c-di-GMP levels than the flat bacteria carpets that grow on rigid substrates, a somehow counter-intuitive result given the paradigm that c-di-GMP production upregulates biofilm-inducing genes, in particular matrix production, while downregulating motility. Here, we describe a case when motility is rendered impossible by the micromechanical properties of the environment rather than by the absence of functional pili, thus resulting in the rapid formation of compact colonies on soft substrates. Further exploring the density and the exact composition of the extracellular matrix in these colonies would be interesting since this parameter could influence the subsequent fate of bacteria on the surface. EPS distribution, composition and concentration may also be significant for the recruitment of new cells on the surface:

indeed, previously deposited matrix is thought to strengthen adhesion of *P.aeruginosa* bacteria (*Zhao et al., 2013*), and could also possibly mediate adhesion of other microorganisms.

In a wider context, the process we observe could also be envisioned as a strategy to optimize bacterial colonization of mechanically heterogenous environments by ensuring accumulation of bacteria into dense colonies located in the softer regions of their environment, for example over cellular tissues. Recently, *Cont et al., 2020* have shown that dense colonies were able to deform soft substrates and exert forces that could disrupt an epithelium layer: rigidity-modulated twitching could thus provide *Pseudomonas aeruginosa* with a convenient means of targeting soft tissues for cooperative disruption and subsequent invasion.

The phenotypic differences that we report for colonies are likely to impact subsequent interactions of bacteria with their environment: response to changes in nutrient or oxygen availability, and chemical signals in general which will not efficiently penetrate inside dense colonies. This could in particular influence susceptibility to antibiotics, as confirmed by very recent work *Cont et al., 2023*. This is all the more relevant that PA can invade many different environments, and might have to be treated differently when it settles in the lungs of cystic fibrosis patients, or on the surface of rigid implants.

Finally, our data show that rigidity-modulated twitching has a striking impact on the mixing of different strains upon surface colonization. Understanding the mechanisms governing the formation of mixed-species communities is one of the key challenges of current biofilm research. Since the motility modulation mechanism described here is quite general and should be marginally affected by the particulars of different strains/species moving through elongation/retraction of an appendage, we expect it to provide a relevant framework to study co-colonization in different mechanical micro-environments.

## Materials and methods

### Bacterial strains

Strains used in this study were *Pseudomonas aeruginosa* wild-type (WT) PAO1, fluorescent strains PAO1 miniCTX-PX2-gfp and PAO1 miniCTX-PX2-eyfp (a kind gift from PBRC group, IBS, Grenoble, unpublished), and PAO1 mutants pilA::Tn5, sadC::Tn5 and wspR::Tn5 obtained from the transposon library at University of Washington (*Jacobs et al., 2003*). Strain PAO1 pCdrA-gfp was obtained by transforming plasmid *pCdrA::gfp^c* from *Rybtke et al., 2012* in our WT strain.

Bacteria were inoculated in Luria-Bertani (LB) medium from glycerol stocks, and grown overnight at 37 °C at 250 rpm. The next morning, 10 µL of the stationary phase culture were diluted in 3 mL of LB medium and placed in a shaking incubator (37 °C, 250 rpm) for 3.5 hr, to reach mid-exponential phase (OD$_{600}$=0.6–0.8). Bacteria were then diluted to OD$_{600}$=0.005 in our working medium, TB:PBS, and inoculated into the channel. TB:PBS is obtained by mixing TB (Tryptone broth, Euromedex, 10 g.L$^{-1}$) and PBS (w/o calcium and magnesium) with a volume ratio of 1:2. We found that this minimal medium favors bacterial twitching for a few hours after adhesion.

### Microfluidic device

Microfluidic channels were cut into 100 µm-thick double-sided sticky tape (7641 W #25, Teraoka, Japan) with a die-cutter. Typically, a 5 cm-long x 1 mm-wide channel was used to bind together a rectangular glass coverslip bearing the hydrogel patches, and a flat 5 mm-thick slab of polydymethylsiloxane (PDMS, Sylgard prepared by mixing crosslinker and monomer solutions 1:10 and baking at 65 °C for 1 hr). Two channels were stuck together to obtain a height of 200 µm, in order for the flow through the channel to not be significantly modified by the 25-µm-thick PAA hydrogels. These sticky-tape channels were first adhered onto the PDMS piece and then placed onto the dehydrated hydrogels. To ensure proper binding, the whole device was placed under vacuum for 30 min. Next, the channel was rinsed with TB:PBS (1:2) for a minimum of 1 hr, in order to rehydrate the hydrogels. Medium was placed in a plastic container and withdrawn into the channel with a syringe pump (Harvard Apparatus, USA, 30 µL/min) to avoid the formation of bubbles.

### Gels and substrates preparation

Hydrogels of polyacrylamide (PAA) and polyethyleneglycol (PEG) were prepared following previously established protocols (*Tse and Engler, 2010*; *Beamish et al., 2010*). All reagents were obtained from

**Table 1.** Hydrogel compositions and associated Young's moduli.

| Acrylamide (wt%) | Bis-acryl. (wt%) | PEGDA (wt%) | Modulus (kPa) |
| --- | --- | --- | --- |
| 4 | 0.225 | 0.0 | 2.7 ± 0.3 |
| 5 | 0.225 | 0.0 | 6.1 ± 0.2 |
| 8 | 0.264 | 0.0 | 18.5 ± 0.7 |
| 20 | 0.47 | 0.0 | 65 ± 5.6 |
| 15 | 0.65 | 0.0 | 84 ± 1.1 |
| 20 | 0.7 | 0.0 | 103 ± 3.8 |
| 0 | 0 | 5 | 5.7 ± 0.3 |
| 0 | 0 | 20 | 102 ± 8.4 |

Sigma Aldrich and used as received: Acrylamide solution (AA, 40% in water), N,N'-Methylenebisacrylamide (Bis, 2% in water), Ammonium Persulfate (APS, $\geq 98\%$), N,N,N',N'- Tetramethylethylenediamine (TEMED, $\geq 99\%$), Poly(ethylene glycol) diacrylate (PEGDA, $M_n \sim 6000$ g.mol$^{-1}$), 2-Hydroxy-4'-(2-hydrox yethoxy)–2-methylpropiophenone (Irgacure 2959, 98%), Bind-silane, Sigmacote.

Rectangular glass coverslips (24 × 60 mm) were used as substrates for gel casting. They were plasma-cleaned and immersed in a solution of Bind-silane (60 µL of Bind-silane, 500 µL of 10% acetic acid, 14.5 mL of ethanol) for 1 hr before being rinsed with ethanol, water, and blow-dried with nitrogen before use. Round glass coverslips (12 mm diameter) were used as counter-surfaces for gel casting. After plasma cleaning, they were immersed in Sigmacote for 1 hr before rinsing with acetone, ethanol and water, and blow-dried before use.

Bulk solutions of AA/Bis and PEGDA were prepared in phosphate buffer saline (PBS) and stored at 4 °C until use. The final stiffness of the gels was tuned by adjusting the AA/Bis or PEGDA content according to *Table 1*. PAA gels were obtained by adding 1 µL of TEMED and 1 µL of a freshly made APS solution (10 w% in water) to a volume of 168 µL of AA/Bis solution. A 3 µL droplet of the mixture was immediately placed on the surface of a bindsilane-treated glass coverslip, sandwiched by a Sigmacote-treated round coverslip, and left for curing for 1 hr in a water vapor-saturated atmosphere. After curing, the round coverslip was lifted off using the tip of a scalpel blade, resulting in a circular pad of gel, of thickness $\sim 25 - 30$ µm, covalently bound to the bottom rectangular coverslip and exposing its free top surface. Circular gel pads were then scrapped with a razor blade in order to adjust their lateral size to the width of the microfluidic channels into which they would eventually be installed. Gel pads were then copiously rinsed with ultrapure water, and left for drying in a laminar flow cabinet. Up to three such pads, with different elastic properties, were prepared simultaneously on the same coverslip, arranged to fit along the length of the microfluidic channel. PEG gels were obtained by adding 5 µL of a 10 wt% solution of Irgacure in ethanol to 0.5 mL of PEGDA solution. A 3 µL droplet of the mixture was placed in between coverslips as described above, and irradiated under UV light (365 nm, 180 mW.cm$^{-2}$) for 15 min for curing. Subsequent steps were as described above for PAA gels.

## Mechanical characterization

The viscoelastic properties of the various gels were characterized by AFM microrheology, using the 'contact force modulation' technique described recently and validated on hydrogels (*Abidine et al., 2015*). It allows determining elastic and loss shear moduli, $G'$ and $G''$, as a function of frequency over the range $1 - 300$ Hz. The Young moduli reported in *Table 1* have been computed as $E = 3G'_0$, with $G'_0$ the low frequency plateau modulus obtained by microrheology, assuming a Poisson ratio $\nu = 0.5$ for all gels. All gel samples displayed elastic behavior with $G' \gg G''$.

Measurements were performed on a JPK Nanowizard II AFM, with pyramidal-tipped MLCT probes (Bruker) of spring constant 15 mN/m. Data were analyzed using a home-written software for microrheology. Thirty-µm-thick gels were prepared, as described above, on round coverslips mounted at the bottom of 35 mm petri dishes. They were then either characterized immediately or left to dry to mimic the protocol used for inclusion in the flow chamber. Experiments were performed in PBS +1 % vol. tween 20 (Sigma), with Tween used to prevent adhesion of the AFM tip to the gel. All

measurements were carried out at 37 °C to mimic experimental conditions with bacteria. Results were compared with force-distance indentation curves that gave consistent results at low rigidities (< 20 kPa) but overestimated the rigidity for higher values (*Figure 1—figure supplement 1a*).

Homogeneity of the gels was assessed at the µm and mm scales by multiposition measurements. We found very good repeatability of the measurements and homogeneity of the gels at all scales (*Figure 1—figure supplement 1b*). Subsequent measurements were hence acquired at three to six different points in the gels and the average and standard error of the mean are provided (*Table 1*). Rigidity was also measured before and after drying and rehydration of the gel to check for possible damage to the structure. In addition, confocal images of the surface of fluorescently labelled gels were used to track default on the gel surface before and after drying. We found no evidence of damage to the hydrogel upon drying, except for very soft gels of rigidity below 1 kPa that were not used in this study (*Figure 1—figure supplement 1c*).

## Microscopy experiments

Diluted bacterial solution was injected into the channel, and kept without flow for 30 min to allow bacteria to attach. During that time, clean tubing was connected to a syringe and filled with TB-PBS medium supplemented with 3 mM glucose and connected to the inlet of the device. 30 min after injecting bacteria into the device, the flow of clean medium was initiated. The flow rate was first set at 25 µl/min for 3 min in order to flush out unattached bacteria, and then lowered to 1 µL/min and maintained constant with a syringe pump (Pico Plus, Harvard Apparatus) throughout the acquisition (yielding a wall shear stress of 2.5 mPa). The set up was immediately placed into the incubation chamber (37 °C) of a Leica SP8 confocal microscope, and acquisition was started at 1 frame/minute.

For matrix staining experiments, concanavalin A (Alexa Fluor 647 conjugate, ThermoFisher Scientific) was added to the medium (3 µl/ml of a 1 mg/ml stock solution) and infused in the flow cell for at least 30 min prior to imaging. Since the tetravalent conA interferes with the structure of the matrix, it was used either for short-term imaging of bacterial twitching at early stages (t<1 h, *Figure 4D*), or added at the end of an acquisition to assess matrix distribution on and around colonies (t $\sim$ 8 h, *Figure 4—figure supplement 1*).

For control experiments in wells (*Figure 1—figure supplement 3*), the protocol was modified as follows: PAA gels were prepared as described above, but at the center of a 35 mm round glass coverslip. The coverslip was then glued (5 min epoxy, Araldite) to the bottom of the well of a six-well plate (Costar, Corning), previously cut-out to open a circular 32 mm hole. One mL of diluted bacteria suspension ($OD_{600}$=0.005) were deposited on the gel, incubated for 30 min, and then carefully pipetted out, and the well was filled with 3 mL fresh medium (TB-PBS +3 mM glucose) and kept at 37 °C. This setup allowed continuous imaging of bacteria on a Zeiss Axio-observer 7 inverted microscope in phase contrast mode (63 x objective) equipped with an Orca-Flash 4.0 LT camera (Hamamatsu).

## Surface coverage analysis and tracking of individual bacteria

All image processing and analysis, unless otherwise noted, was performed with Fiji using available plugins and home-written macros. In order to quantify the movements of individual bacteria, time series of phase-contrast images were registered using the Fiji plugin "MultiStackReg" *Thévenaz et al., 1998* and segmented with the plugin "weka trainable segmentation" *Arganda-Carreras et al., 2017*. The resulting segmentation was checked and corrected manually.

For the analysis of the global velocity $V_g$, segmented binary images were used to estimate the surface coverage $A(t)$ and the explored surface area $S(t)$, using a home-written MATLAB script. Briefly, at each timepoint 2 binary images were generated: one where pixels occupied by bacteria were assigned the value 1, and all others zero (providing $A(t)$), and another image obtained by adding all binary images up to this timepoint, so that all visited pixels were assigned the value 1 (providing $S(t)$).

For the analysis of individual displacement steps, segmented bacteria were fitted with an ellipse, and the 'analyze particle' imageJ function was subsequently used to locate the center of mass of each bacterium. The Fiji plugin 'TrackMate' *Tinevez et al., 2017* was used to track all individual bacteria, again followed by manual validation and correction (see *Video 2*). The function importTrackMateTracks (*Tinevez et al., 2017*, https://github.com/fiji/TrackMate/blob/master/scripts/importTrackMateTracks.m) was used to import tracking data into MATLAB, and homemade scripts were used to sort data, plot tracks and obtain velocity distributions.

The analysis of the histograms of displacement steps were performed as follows: we assume that the measured steps are the incoherent sum of two displacement vectors, the active displacement $\vec{V}_a$ due to T4P activity, and a vector $\vec{V}_p$ that includes passive effects due to both the noise on measurements, and displacements resulting from bacterial growth and local crowding. First, we considered the experimental distribution obtained with the *pilA* mutant, for which $\vec{V}_a = \vec{0}$ : this allows extracting the probability distribution for $\|\vec{V}_p\|$, which can be well fitted with a decreasing exponential with a characteristic passive velocity $V_{C,p}$ : $p(\left\|\vec{V}_p\right\|) = exp(-\frac{\|\vec{V}_p\|}{V_{C,p}})$, with $V_{C,p} = 0.044\ \mu m/min$. We then considered the case of twitching bacteria. Here, we observed that the tail of the displacement step distribution also follows a decreasing exponential trend. Based on the reasoning that passive displacements are short-ranged and should not significantly modify the distribution for large displacement values, we deduce that the tail of the probability distribution for $\|\vec{V}_a\|$ is a decreasing exponential, $p(\|\vec{V}_a\|) = exp(-\frac{\|\vec{V}_a\|}{V_C})$, with $V_C$ the characteristic active twitching velocity of bacteria.

We confirmed the validity of this hypothesis by calculating the probability distribution functions. We assumed that the distribution of measured displacement steps, $\|\vec{V}_{tot}\| = \|\vec{V}_a + \vec{V}_p\|$ is the sum of two uncorrelated exponential distributions with different scales and a random angle between the two vectors. There is no analytical expression for this sum, hence we performed numerical calculations of the distributions obtained in the general case. In the limit $\|\vec{V}_{tot}\| >> V_C > V_{C,p}$ an exponential distribution is retrieved with a characteristic velocity $V_C$, unaffected by $V_{C,p}$ (*Figure 3—figure supplement 3*, left). A fitting of the range $p(\|\vec{V}_{tot}\|) < 0.3$ (which excludes the first few points that do not follow an exponential trend) confirms that $V_C$ is obtained accurately provided that $V_C > V_{C,p}$ (*Figure 3—figure supplement 3*, right). Below this limit, only $V_{C,p}$ is detected since active displacements are in the range of passive 'noisy' ones.

Experimentally, we have used a lower cutoff of $\|\vec{V}_{tot}\| > 0.08\ \mu m/min \simeq 2V_{C,p}$ for the fitting range, to restrict it to the exponential part of the distribution. To account for the noise in the measurement, we have also considered that fitted values below $V_{C,p} = 0.044\mu m/min$ were in the range [0;0.044] m/min.

## Quantification of the morphology of colonies

Quantification was performed on confocal fluorescence 3D resolved images. First, signal attenuation with depth was compensated by decreasing exponential fitting of the mean pixel values inside the colony with depth, and normalization by the corresponding function. A 2D 3x3 smoothing operation was then performed on each image of the z-stack, and the colonies were subsequently segmented using a simple thresholding operation: while this procedure does not permit segmentation of individual bacteria, it provides a good estimate of the 3D envelope of the colonies. The topology of the colonies was then quantified by calculating the roughness of this envelope using the widely used arithmetic average roughness $Ra$

$$Ra = \frac{1}{N}\sum_{i=1}^{N}|z_i - \langle z \rangle|, \tag{12}$$

where summation is over all 2D positions $i$ in the 3D image, $z_i$ is the height of the highest segmented pixel at position $i$ and $\langle \rangle$ is the averaging operator over all positions. The occupied volume $V$ is calculated as

$$V = px^2\sum_{i=1}^{N}z_i, \tag{13}$$

with $px$ the pixel size. The occupied area as a function from the distance to the coverslip is the histogram of $z_i$ values with bin size 0.5 µm (corresponding to the vertical sampling of the 3D images).

## Quantification of the mixing of two strains co-colonizing the same soft substrate, as a function of the softness

This quantification is performed both at the low density stage with isolated bacteria, and at a later stage on maturing colonies. To this aim, we used a statistical tool, Moran's I index, designed to quantify the spatial clustering of species and widely used in the field of ecology and geography *Moran, 1950*. Moran's I is a measure of the local spatial correlations that includes a notion of spatial proximity,

either in the form of a spatial cut-off for the calculation of the heterogeneity (in other words, a characteristic distance), or a number of neighbors. It takes values ranging from 1 (perfectly correlated values) to –1 (perfectly anti-correlated values), with 0 corresponding to a spatially random distribution of the variable.

Considering a variable $y$ that can take two different values (in our case, green (1) or yellow (–1)) with $n$ realisations, Moran'I is expressed as:

$$I = \frac{n}{\sum\limits_{i=1}^{n}\sum\limits_{j=1}^{n} w_{ij}} \frac{\sum\limits_{i=1}^{n}\sum\limits_{j=1}^{n} w_{ij}\left(y_i - \langle y \rangle\right)\left(y_j - \langle y \rangle\right)}{\sum\limits_{i=1}^{n}\left(y_i - \langle y \rangle\right)^2}, \tag{14}$$

where $w_{ij}$ is the matrix of weights that contains the spatial information (with $w_{ii} = 0$). In our experiment, the relevant spatial scale (and hence the matrix of weights) varies greatly over time because of the change in the density of the bacteria on the surface. While at high density (maturing colonies) defining a length scale is a suitable way of testing the presence of local correlations, this is more challenging at earlier times when the distance between neighbors exhibits large stochastic variations, in particular for stiff substrates. Hence, different matrices of weights were chosen for early-stage and later-stage colonization of the surface:

- At early stages of colonization, when the bacteria are sparse on the surface, we chose to focus on the nearest neighbors of each bacteria. To this aim, individual bacteria are segmented in the green and yellow images, and their center of mass location is collated into a list of 2D coordinates and colour for all bacteria in the field of view. Moran's I is then calculated based on this list using the following weight matrix:
  We arbitrarily chose $p = 5$ as a significant number of neighbors, although similar results are found for values of $p$ ranging from 4 to 10. Lower numbers are biased by cell division: at the time of division, the closest neighbor is necessarily of the same strain as the bacteria under consideration, so that there is always a positive correlation between them. As a result, testing for mixing requires to mitigate this effect by choosing a large enough value for $p$. In practice, we found that $p = 5$ was a good compromise to limit this bias while maintaining a 'local' approach, that is not considering the correlation between bacteria further apart than half of the field of view (i.e. 160 μm).
  - $w_{ij} = \begin{cases} 1 & \text{if } j \text{ is one of the } p \text{ nearest neighbours of bacteria } i \\ 0 & \text{otherwise} \end{cases}$
- At later stages with dense, 3D colonies, individual segmentation of bacteria becomes challenging and the correlation measure is performed on individual pixels: first, a simple thresholding operation is performed on the green and yellow image, and each pixel is attributed a value: 1 (green pixel), 0 (black pixel) or –1 (yellow pixel). From this new image, Moran's I is calculated using the following weight matrix:
  Again, the cut-off distance $d$ is arbitrarily chosen as 5 μm although values between 3 and 10 μm yield similar results: it permits limiting fluctuations by averaging over a significant number of bacteria, while maintaining a local measure of mixing. In addition, because individual bacteria cover more than one pixel in the acquired images, a number of pixels of the same colour as pixel $i$ are removed to avoid correlating the bacteria with itself. In our data the average number of pixels covered by one bacteria is measured to be 40.
  - $w_{ij} = \begin{cases} 1 & \text{if the distance between } i \text{ and } j \ (i \neq j) \text{ is smaller than or equal to } d \\ 0 & \text{otherwise} \end{cases}$

While there is some degree of freedom on the choice of the weight matrix, it is important to note that we use the same weight to compare data obtained on three different rigidities, hence minimising the impact of the exact chosen parameters on the comparison. In contrast, values obtained on one surface at the two different time points should not be directly compared as they have not been obtained with the same weight matrix.

## Acknowledgements

We thank Claude Verdier for help with the AFM elasticity measurements, Denis Bartolo for help with microfluidics, and Benoit Coasne and Benedikt Sabass for fruitful discussions on data modeling. We are extremely grateful to Ina Attree and Sylvie Elsen (IBS, Grenoble) for strains, help and advice. We thank Tim Tolker Nielsen for the kind gift of pcdrA-gfp reporter plasmids. DD was supported by the French National Research Agency (grant ANR-19-CE42-0010). The authors acknowledge support from LabeX Tec 21 (ANR-11-LABX-0030).

## Additional information

### Funding

| Funder | Grant reference number | Author |
| --- | --- | --- |
| Agence Nationale de la Recherche | ANR-19-CE42-0010 | Delphine Débarre |
| Labex Tec21 | ANR-11-LABX-0030 | Lionel Bureau<br>Karin John<br>Delphine Débarre<br>Sigolene Lecuyer |

The funders had no role in study design, data collection and interpretation, or the decision to submit the work for publication.

### Author contributions

Sofia Gomez, Validation, Investigation, Methodology; Lionel Bureau, Conceptualization, Data curation, Formal analysis, Supervision, Validation, Investigation, Writing - review and editing; Karin John, Formal analysis, Methodology, Writing - review and editing; Elise-Noëlle Chêne, Investigation, Methodology; Delphine Débarre, Sigolene Lecuyer, Conceptualization, Data curation, Formal analysis, Supervision, Validation, Investigation, Methodology, Writing - original draft, Writing - review and editing

### Author ORCIDs

Karin John http://orcid.org/0000-0003-1678-6880
Delphine Débarre http://orcid.org/0000-0002-0513-6172
Sigolene Lecuyer http://orcid.org/0000-0001-7393-2667

### Decision letter and Author response

Decision letter https://doi.org/10.7554/eLife.81112.sa1
Author response https://doi.org/10.7554/eLife.81112.sa2

## Additional files

### Supplementary files

• MDAR checklist

### Data availability

Figure 2—source data and Figure 3—source data contain the numerical data used to generate the figures.

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

## Appendix 1

## Modeling twitching velocity on soft substrates

The principle of our modeling of rigidity-modulated twitching in 1D is shown in *Figure 2D* (main text), and incorporates three main ingredients: modeling of the substrate deformation (subsection A), of cell body friction on the surface (subsection B) and of the pilus retraction dynamics (subsections C and D).

## A. Modeling substrate deformation

We have based our approach on the theory of linear elasticity for the description of the substrate: in this framework, the deformation of the substrate occurs over a typical length scale given by the size of the adhesion, $\lambda$, and it is proportional to the applied force. Finally, the proportionality coefficient $Y$ scales as the product of the substrate elastic modulus, $E$, and the adhesion size $\lambda$, that is $Y = E\lambda$. This simple relation is valid only for small displacements on rigid substrates. It is likely to fail quantitatively on very soft substrates with large displacements, low cross-linker densities and non-affine deformations, but is a reasonable first approximation for the simple model we propose here.

This modeling introduces characteristic length scales that depend on the part of the bacteria under consideration: both the pilus and the cell body form contact with the substrate. The pilus attaches at its tip over size $\lambda \approx 1$ nm, while the cell body has a typical size of $l_b \approx 1$ μm. In addition, a third length scale is the typical length of the pilus, $L$, which varies during retraction but is most of the time $> 1$ μm. Introducing these three quantities permits to simplify the description of the deformation of the substrate: the pilus tension $F$ and the displacement at the adhesion site in the substrate $u$ are linearly related by $F = Yu$, with $Y$ being an effective spring constant. We model the substrate as an infinite (thickness $\approx 25$ μm $\gg \lambda$, lateral extension $\approx 1 - 10$ mm $\gg \lambda$), isotropic, elastic and incompressible half space. Furthermore, we neglect the influence of the cell body on the deformation around the pilus tip since $L >> \lambda$ so that the deformation of the substrate has decayed to zero at the cell body.

The 2D Boussinesq Green's tensor at the surface $z = 0$ for a point-like shear force $\mathbf{f}$ at the origin is given by *Landau and Lifschitz, 2004*

$$\mathbf{G}(\mathbf{r}) = \frac{3}{4\pi E} \left[ \frac{\mathbf{I}}{r} + \frac{\mathbf{r} \otimes \mathbf{r}}{r^3} \right] \mathbf{f}. \tag{A1}$$

Considering an adhesive T4P tip of length $\lambda$ and half-width $d$ and using slender body approximations, the total force $F$ on the pilus for a "lengthside" displacement $u$ is given by

$$F = \frac{E\lambda\pi}{3 \ln \frac{\lambda}{d}} u \quad \text{with} \quad Y = \frac{E\lambda\pi}{3 \ln \frac{\lambda}{d}} \approx E\lambda. \tag{A2}$$

Here we have implicitly introduced a 1D setting, that is we will neglect the vectorial nature of forces and displacements and restrict ourselves to a 1D setting. We find, as expected, that $Y$ scales linearly with $\lambda$. This holds equivalently for the cell body by replacing $\lambda$ with $l_b >> \lambda$: as a result, the substrate deformation at the cell body caused by the same pilus tension $F$ is of amplitude smaller by a factor $\lambda/l_b \ll 1$ and will be neglected for the sake of simplicity.

In contrast, we consider the pilus tip to be firmly attached to the substrate until detachment while the cell body can slide on the surface. Note that this asymmetry between bacterial body (macroscopic sliding over the substrate) and the supposedly small pilus/substrate contact (point-like force deforming the substrate) is the essential difference to the pulling process described in *Simsek et al., 2019*, where the contact of the bacterial body and the pilus extremity are mechanically treated as equivalent.

## B. Modeling cell body friction

As stated above, the model requires a description of the sliding motion of the cell body on the substrate as a function of the force $F$ applied by the pilus. We base our modeling on the theory from *Sens, 2013* that considers stochastic friction by an ensemble of N elastic linkers (not necessarily all bound at all times) between an elastic substrate and a cell, submitted to a sliding velocity $v$. The

bonds are modeled as slip bonds with a critical force $f^*$, an off-rate constant at zero force $k_{\text{off}}^0$ and an on-rate constant $k_{\text{on}}$. The linkers' stiffness is $k_{\text{b}}$.

In the case of an infinitely rigid substrate, the mean total force on the cell body $\langle F \rangle$ as a function of its velocity $v$ is non-monotonous and is given by

$$\langle F \rangle = Nf^* \frac{r_{\text{on}}\, e^{1/\tilde{v}} \int_0^\infty f e^{-\left(\frac{e\,f}{\tilde{v}}\right)}\, \mathrm{d}f}{\tilde{v} + r_{\text{on}}\, e^{\left(\frac{1}{\tilde{v}}\right)}\, \Gamma\left[0, \frac{1}{\tilde{v}}\right]} \tag{A3}$$

with $\tilde{v} = v/v_\beta$, $v_\beta = k_{\text{off}}^0 f^*/k_{\text{b}}$, $r_{\text{on}} = k_{\text{on}}/k_{\text{off}}^0$. $\Gamma[0, x]$ is the Euler gamma function. *Equation (A3)* exhibits a complex dependence of $\langle F \rangle$ on $\tilde{v}$ that requires estimating typical values of the parameters in our experiments. Putting in numbers to obtain the typical speed $v_\beta$, we can estimate that

- $k_{\text{off}}^0 \approx 1 - 10\ s^{-1}$ (slightly higher than for specific ligand/receptor bonds *Robert et al., 2007*)
- $f^* = k_{\text{B}}T/x_\beta$ with $x_\beta \approx 0.1 - 10$ nm being the transition state distance between bound and unbound state as proposed by *Evans, 2001* and others *Pereverzev et al., 2005*.
- $k_{\text{b}}$ is more difficult to estimate. Here we assume that bacterial adhesion is mediated by the bacteria produced extracellular matrix, of which a major constituent are exopolysaccharides. Using a worm-like chain (WLC) model for a polymer of persistence length $L_{\text{p}} \approx 10$ nm (as calculated for bacteria produced exopolysaccharides in *Kuik et al., 2006*) and contour length $L_0 \approx 100$ nm (assuming a chain length of about 100 monomers with size 1 nm), the linear force-elongation relationship in the regime of weak forces *Marko and Siggia, 1995* yields a force constant $k_b \approx \frac{3kT}{2L_{\text{p}}L_0} \approx 6 \times 10^{-3}$ pN.nm⁻¹.

Taking extreme values this leads to typical velocities in the range $v_\beta = 1 - 100$ µm.s⁻¹. In our experiments the bacteria are not expected to move faster than the pilus retraction velocity (i.e. 1 µm.s⁻¹ Skerker and *Skerker and Berg, 2001*, if one excepts the case of slingshots that were not frequently observed in our experiments). Taking into account that the pilus retraction speed slows down considerably as the tension in the pilus increases, the bacterial speed during one pilus retraction is rather smaller than this maximum value. Hence, we always have $\tilde{v} = v/v_\beta < 1$, and *Equation (A3)* can be linearized to

$$\langle F \rangle = Nf^* \frac{k_{\text{on}}}{k_{\text{off}}^0 + k_{\text{on}}} \tilde{v}, \tag{A4}$$

In addition, the elasticity of the substrate should be considered. *Sens, 2013* proposes that this situation is equivalent to having a system of springs in series, one stemming from the substrate elasticity and the second being the collection of individual bond springs (in parallel). In this case and using once again the theory of linear elasticity, the previous analysis holds if $\tilde{v}$ is rescaled by a factor $\frac{E\,l_b}{k_b + E\,l_b}$, with $E > 3$ kPa the substrate Young's modulus and $l_b \approx 1$ µm the characteristic size of the bacterial cell body, $a = l_b \approx 1$ µm. Hence $E l_b \geq 3$ pN.nm⁻¹ $\gg k_b$ and the scaling factor $\frac{E\,l_b}{k_b + E\,l_b} \approx 1$, so that the elasticity of the substrate does not influence the friction of the cell body. In summary, we find that we can reasonably use a linear approximation for the bacterial sliding speed in response to the pulling force due to the pilus retraction, $F = \eta v$ with $\eta$ a friction coefficient. Finally, we consider $\eta$ as independent from the substrate rigidity, which is reasonable if we assume that the number of bonds is limited by the number of molecules/appendages of the cell body that can interact with the substrate, rather than the number of binding sites on the substrate itself (PAA mesh size $\approx 3 - 10$ nm), and that the interaction may in addition be mediated by adsorbed exopolysaccharides deposited by the bacteria. However, other non-linear dependencies can be easily included into the modeling.

## C. Basic modeling of pilus retraction

The relevant step during twitching which induces bacterial motion is the active pilus retraction when attached to the substrate. Here we assume, that the limiting effect for bacterial motion is the detachment of the pilus from the substrate, and not the complete retraction of the pilus by the bacterium. To understand the role in substrate rigidity on the bacterial twitching speed we will therefore concentrate on this crucial step without describing the whole cycle of pilus dynamics, for which the kinetics is not completely understood *Koch et al., 2021*; *Talà et al., 2019*.

We consider the retraction of a single effective pilus pulling on the bacterial body until it detaches from the substrate. We treat the pilus as rigid and inextensible filament: assuming a force constant of 2 pN.µm⁻¹ for the pilus elasticity *Beaussart et al., 2014*, a substrate rigidity of $E = 100$ kPa, an

adhesion size of $\lambda = 1$ nm and a maximum force exerted by the pilus of $F_R = 100$ pN, the substrate displacement is $u \sim F_R/E\lambda = 1$ µm. In contrast, the pilus elongation is $\Delta L = 50$ nm and can therefore be neglected for our conditions. However it would not pose any difficulty to include the pilus elasticity into the calculations.

Let $v_R$ be the retraction speed of the attached pilus inducing a displacement $u$ in the substrate. At the same time the bacterium will slide forward with speed $v_B$, reducing the tension in the pilus and the displacement in the substrate:

$$\frac{\mathrm{d}u}{\mathrm{d}t} = v_R(F) - v_B(F) \quad \text{with} \quad F = Yu\,. \tag{A5}$$

Both motions (substrate displacement and bacterial sliding) are coupled via the tension in the pilus $F$. Its retraction speed is described by a simple linear dependence that has been well documented *Marathe et al., 2014*; *Koch et al., 2022*

$$v_R = v^0 \left(1 - \frac{F}{F_R}\right)\,, \tag{A6}$$

with $F_R$ a stall force. As established in the previous subsection, the bacterial sliding speed depends linearly on the pilus tension with friction constant $\eta = F_B/v^0$:

$$v_B = \frac{1}{\eta}F = v^0 \frac{F}{F_B}\,. \tag{A7}$$

$F_B$ denotes the force necessary to pull the bacterium at maximum retraction speed $v^0$ over the substrate. From *Equation A5* we recover the increase in the pilus tension over time during the retraction

$$F(t) = F_0 \left(1 - e^{-\frac{Yv^0}{F_0}t}\right) \tag{A8}$$

with the force scale

$$F_0 = \frac{F_B F_R}{F_R + F_B}\,. \tag{A9}$$

Incorporating solution (A8) into *Equation (A7)* with $v_B = \frac{\mathrm{d}x_B}{\mathrm{d}t}$ we recover for the bacterial sliding distance during pilus retraction

$$x_B(t) = \frac{F_0}{F_B} \left[v^0 t + \frac{F_0}{Y} \left(e^{-\frac{Yv^0}{F_0}t} - 1\right)\right]\,. \tag{A10}$$

While retracting the pilus will detach with a rate constant $k_{\mathrm{off}}(F)$ from the substrate. Assuming a force independent off-rate constant $k_{\mathrm{off}} = k_{\mathrm{off}}^0$ the detachment times are distributed exponentially with mean $1/k_{\mathrm{off}}^0$. Furthermore, we assume that the single effective pilus considered in our model retracts with frequency $k_p$ and thus gives rise to an effective velocity

$$v_{\mathrm{eff}} = k_p \langle x_B \rangle = k_p k_{\mathrm{off}}^0 \int_0^\infty x_B(t)\, e^{-k_{\mathrm{off}}^0 t}\, \mathrm{d}t = V_{\mathrm{max}} \frac{E}{E + E_0}\,. \tag{A11}$$

Here, $\langle x_B \rangle$ denotes the mean bacterial sliding distance per pilus retraction event. $V_{\mathrm{max}}$ denotes the maximum effective speed that a cell can reach on a given substrate at infinite rigidity, given by

$$V_{\mathrm{max}} = v^0 \frac{k_p}{k_{\mathrm{off}}^0} \frac{F_R}{F_B + F_R}\,. \tag{A12}$$

$E_0$ denotes the rigidity at half-maximal speed and is given by

$$E_0 = \frac{F_B F_R k_{\mathrm{off}}^0}{(F_B + F_R)v^0 \lambda}\,. \tag{A13}$$

*Appendix 1—figure 1* shows the experimental data and fitted curves, which capture well the data for medium and high rigidities. The theoretical curves were fitted to all experimental values (applying the statistical weight in the measured rigidities and equal weight in the velocities) using a least square fit (software gnuplot *Williams and Kelley, 2019*). Assuming a typical pilus retraction speed $v^0 = 0.5 - 1$ µm.s$^{-1}$ *Marathe et al., 2014*; *Koch et al., 2022*, a stall force of the order $F_R = 50 - 100$ pN *Marathe et al., 2014*; *Koch et al., 2022*, a pilus off-rate constant $k_{off}^0 = 1$ s$^{-1}$ *Talà et al., 2019* and a contact size of $\lambda = 1$ nm *Koch et al., 2022*, a high friction surface with $F_B = 1$ nN and a typical pilus retraction frequency (Here we assume that one single effective pilus is active during a retraction event. Using a typical pilus length of 5 µm with retraction speed of $v_0 = 0.5 - 1$ µm.s$^{-1}$ gives a duration of 5–10 s per retraction and a retraction frequency of 0.1–0.2 s$^{-1}$) of $k_p = 0.1 - 0.2$ s$^{-1}$ we recover a $V_{max} \sim 0.1 - 1$ µm.min$^{-1}$ and a substrate rigidity at half maximum speed of $E_0 = 10 - 100$ kPa, a range which encloses the fitted values (see *Appendix 1—figure 1*).

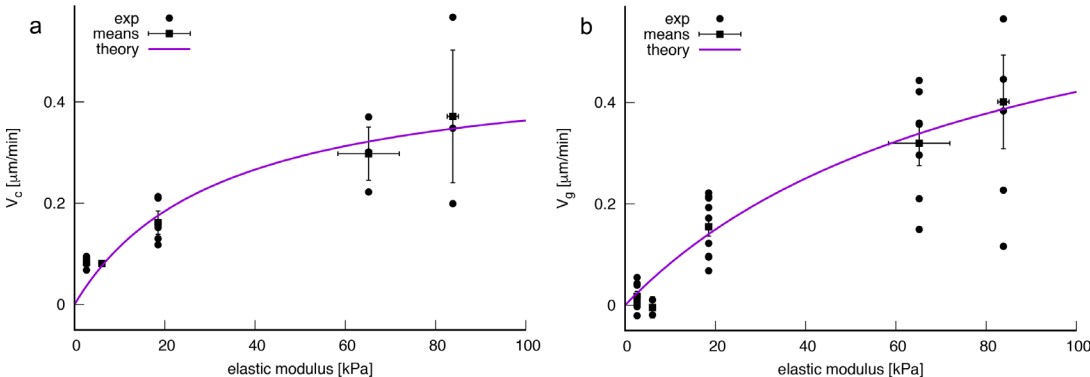

**Appendix 1—figure 1.** Experimentally measured velocity vs rigidity data and least squared fits of *Equation A11* w.r.t. to the experimental values as indicated in the legends. (**a**) Local velocity measures. Parameters obtained by a least-square fit: $E_0 = 32 \pm 20$ kPa, $V_{max} = 0.48 \pm 0.12$ µm.min$^{-1}$. (**b**) Global velocity measures. Parameters obtained by a least square fit: $E_0 = 84 \pm 68$ kPa, $V_{max} = 0.77 \pm 0.35$ µm.min$^{-1}$. Errorbars indicate SEM.

Here we have assumed a force-independent off-rate constant for the pilus. In a more complex scenario, the contact between the pilus and the substrate may act as a slip bond or catch bond. For completeness we will show some numerical results for slip and catch bond behavior below, which do not increase however the quality of fit between experimental and theoretical velocity data.

## D. Force dependent detachment rate constants

Increasing the complexity of the model, we assume that the pilus detachment rate is force dependent *Kramers, 1940*; *Björnham and Axner, 2010*; *Pereverzev et al., 2005*; *Talà et al., 2019* and takes the form

$$k_{off} = k_{off}^0 \left( \varepsilon e^{-\frac{F}{F_C}} + e^{\frac{F}{F_S}} \right) . \tag{A14}$$

$\varepsilon = 0$ denotes a slip bond and $\varepsilon > 0$ denotes a catch bond behavior. $F_C$ and $F_S$ denote positive force constants *Pereverzev et al., 2005*. *Equation (A14)* implies that the pilus detachment times are not distributed exponentially.

We now consider the evolution equation for the probability density $p(u)$ that a pilus attached to the substrate is retracting and is thereby inducing a displacement $u$

$$\partial_t p = -k_{off}(F)p - \partial_u j_u \tag{A15}$$

The first term denotes (tension dependent) pilus detachment from the substrate and the second term captures the advection of the displacement due to pilus retraction and bacterial sliding. It is formulated as a divergence of a flux $j_u$ with

$$j_u = \left[ v_R(F) - v_B(F) \right] p . \tag{A16}$$

The pilus retraction $v_R(F)$ and bacterial sliding speed $v_B(F)$ is given by *Equations (A6) and (A7)*. To facilitate the analysis of the equations we use the transformation $p(u) = p(u[F]) = P(F)$ and $\partial_u = Y\partial_F$ which gives rise to the evolution equation

$$\partial_t P(F) = k_{\text{off}}(F)P - v^0 Y \partial_F \left[ \left( 1 - \frac{F}{F_R} - \frac{F}{F_B} \right) P \right] \tag{A17}$$

To reduce the number of parameters we introduce the timescale $t_0 = 1/k_{\text{off}}^0$, the length scale $l_0 = v^0 t_0$ and the force scale $F_0 = (F_R F_B)/(F_R + F_B)$. The adimensional quantities are then denoted $\tilde{F} = F/F_0$, $\tilde{t} = t/t_0$, and $\tilde{u} = u/l_0$. The adimensional evolution equation of $\tilde{P}(\tilde{F})$ takes the form

$$\partial_t \tilde{P} = -\kappa(\tilde{F})\tilde{P} - \mu \partial_{\tilde{F}} \left[ \tilde{P}(1 - \tilde{F}) \right] , \quad \text{with} \quad \tilde{F} \in [0, 1] \tag{A18}$$

where $\mu = Y v^0/(F_0 k_{\text{off}}^0)$ denotes the adimensional substrate rigidity and $\kappa$ denotes an adimensional force dependent off-rate, that is $\kappa = k_{\text{off}}/k_{\text{off}}^0$. Solving *Equation (A18)* in the steady state we find

$$\tilde{P} = \frac{\tilde{P}_0}{1 - \tilde{F}} e^{\frac{\mathcal{I}(\tilde{F})}{\mu}} \tag{A19}$$

with

$$\mathcal{I}(\tilde{F}) = \varepsilon e^{-\frac{1}{\tilde{F}_C}} \text{Ei}\left( \frac{1 - \tilde{F}}{\tilde{F}_C} \right) + e^{\frac{1}{\tilde{F}_S}} \text{Ei}\left( -\frac{1 - \tilde{F}}{\tilde{F}_S} \right) . \tag{A20}$$

In *Equation (A20)* the force constants $\tilde{F}_C$ and $\tilde{F}_S$ have been rescaled by $F_0$. The normalization factor $P_0$ is defined by the integral condition $\int_0^1 \tilde{P}(\tilde{F}) \, d\tilde{F} = 1$. $\text{Ei}(x)$ denotes the exponential integral. At detachment the distribution of forces $\tilde{F}_d$ is given by

$$\tilde{P}_d(\tilde{F}_d) = \tilde{P}_{d0} \frac{\kappa(\tilde{F}_d)}{1 - \tilde{F}_d} e^{\frac{\mathcal{I}(\tilde{F}_d)}{\mu}} \tag{A21}$$

with the normalization factor $\tilde{P}_{d0}$ determined by the integral condition $\int_0^1 \tilde{P}_d(\tilde{F}_d) \, d\tilde{F}_d = 1$.

Using the forces and bacterial sliding distance at detachment from the substrate

$$\tilde{F}_d = 1 - e^{-\mu \tilde{t}_d} \tag{A22}$$

$$\tilde{x}_B = \frac{1}{\tilde{F}_B} \left[ \tilde{t}_d + \frac{1}{\mu} \left( e^{-\mu \tilde{t}_d} - 1 \right) \right] \tag{A23}$$

we can perform the transformation $\tilde{P}_d(\tilde{F}_d) \, d\tilde{F}_d = \tilde{P}_d[\tilde{F}_d(\tilde{t}_d)] \mu e^{-\mu \tilde{t}_d} \, d\tilde{t}_d = \tilde{P}_{\tilde{t}_d}(\tilde{t}_d) \, d\tilde{t}_d$ and recover the mean bacterial displacement per pilus retraction in adimensional form as

$$\langle \tilde{x}_B \rangle = \int_0^\infty \tilde{x}_B(\tilde{t}_d) \tilde{P}_{\tilde{t}_d} \, d\tilde{t}_d . \tag{A24}$$

Following the same argument as for *Equation A11*, the effective bacterial speed (dimensional) is then given by

$$v_{\text{eff}} = k_p l_0 \langle \tilde{x}_B \rangle = \frac{k_p}{k_{\text{off}}^0} v^0 \langle \tilde{x}_B \rangle \tag{A25}$$

$$= V_{\text{max}} \int_0^\infty \left[ \tilde{t}_d + \frac{1}{\mu} \left( e^{-\mu \tilde{t}_d} - 1 \right) \right] \tilde{P}(\tilde{t}_d) \, d\tilde{t}_d , \tag{A26}$$

with $\mu = E/E_0$.

*Appendix 1—figure 2* shows exemplarily the off-rate constants for force independent, slip and catch bond behavior (*Appendix 1—figure 2a*) and the effective velocity of a slip-bond and catch-bond model along with a force independent detachment in comparison to the measured bacterial velocity using the local velocity analysis (*Appendix 1—figure 2b*). Thereby we chose arbitrarily a slip-bond constant $F_S = 1.1 F_0$ corresponding for example to the case of a high friction substrate

with $F_R = F_S = F_B/10$, that is as used previously $F_R = F_S = 100$ pN and $F_B = 1$ nN. The catch-bond force constant was chosen to be small, that is $F_S \ll F_0$, following the idea of *Talà et al., 2019* that pilus-substrate attachment is stabilized for small pilus tension. Futhermore, we chose $\varepsilon = 2$, that is pilus detachment at zero loads is three times faster than for a slip-bond model. Fixing $E_0 = 32$ kPa [obtained from fitting the force-independent model (see *Appendix 1—figure 1a*)]., the theoretical curves with the force-dependent off-rate constant were fitted using a least square fit in the parameter $V_{\max}$. The catch-bond behavior captures qualitatively better the velocities at low rigidities but neither slip-bond nor catch-bond seem to perform better than the simple analytical force-independent detachment model for medium and high rigidities.

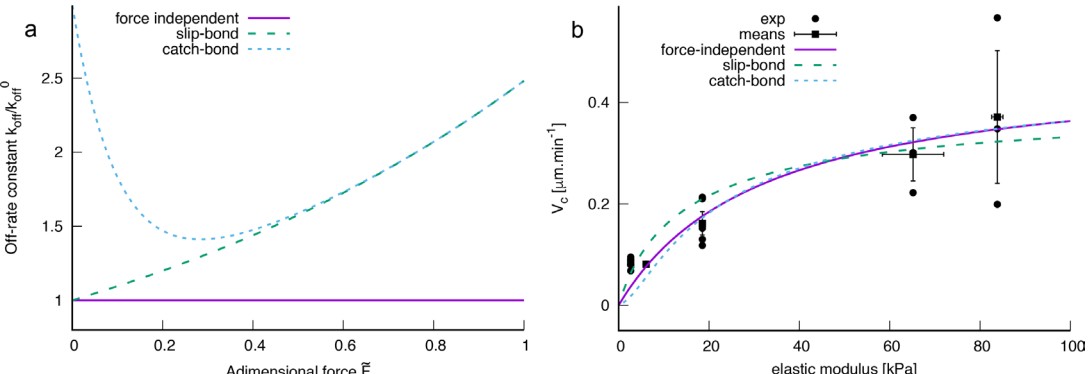

**Appendix 1—figure 2.** Comparison of various force dependencies of the pilus detachment rate constant $k_{off}$ as indicated in the legend. (**b**) Comparison of bacterial velocities obtained by models with various complexity with experimentally measured values (local velocity analysis) as indicated in the legend (parameters were fit w.r.t experimental mean values). The model parameters for the force-independent model are as in *Appendix 1—figure 1a*. For the slip and catch bond model the fixed parameters are $E_0 = 32$ kPa,, $\tilde{F}_S = 1.1$ $\varepsilon = 2$ (catch bond), $\tilde{F}_C = 0.1$ (catch bond). For the slip and catch bond model $V_{\max}$ was estimated from least square fits: $V_{\max} = 0.98$ µm.min⁻¹ (slip) and $V_{\max} = 1.15$ µm.min⁻¹ (catch). Error estimates are expected to be of the same order of magnitude as for the force independent model (see *Appendix 1—figure 1a*).

## Appendix 2

## Influence of bacterial motility on the onset of biofilm verticalization

### A. Simple kinetic model

As described in the main text, we propose a simple kinetic model to capture the 2D to 3D transition of bacteria in growing microcolonies over time, that is we assume that colony verticalization results from a competition between bacterial division and motility, rather than from a competition between adhesion forces between bacteria or between bacteria and the substrate. We thus assume that there is no strong difference in the binding energy of a cell to the substrate as a function of its rigidity, and that this energy is slightly higher than that of binding to another cell. Bacteria thus prefer adhering to the substrate in all cases but can easily adhere to other cells if needed. Based on this assumption, we consider two key features of surface colonization to describe the 2D to 3D transition:

- Growth: initially, at time $t = 0$, one bacterium is attached to the surface. The number of bacteria $N$ grows exponentially with time as:

$$N(t) = e^{\frac{t}{t_0}} .$$ (A27)

The characteristic time scale, $t_0$, accounts both for the growth and for the occasional detachment of bacteria from the surface. De novo attachment of bacteria to the surface is neglected. Furthermore, $t_0$ is assumed to be constant over time and across the different surfaces.

- Movement: bacteria explore the surface with a characteristic velocity $V_{CM}$ and perform a random walk (we consider time and length scales larger than the persistence length/time of bacterial twitching motion). These displacements result in a spreading of the colony over a characteristic area $a(t)$ following a diffusive process:

$$a(t) = a_0 + \alpha V_{CM} t$$ (A28)

where $a_0$ is the area of one bacterium and $\alpha$ is a phenomenological parameter related to the properties of the random walk.

From the two equations above, it is clear that the number of bacteria attached to the surface grows faster than the size of the corresponding colony. Therefore, at a critical time $t_c$ corresponding to a critical number of bacteria $N_c$ on the substrate, the area available to bacteria for spreading on the surface will be completely occupied. that is $a_0 N_c = a_0 N(t_c) = a(t_c)$ and thus

$$N_c = 1 + \gamma V_{CM} \ln(N_c)$$ (A29)

where we have substituted $\ln(N_c)$ for $t_c$ on the *r.h.s.* of **Equation A27** and $\gamma = \alpha t_0/a_0$. Solving this equation permits to obtain $N_c$ as a function of $V_{CM}$ and one unknown parameter, $\gamma$. Note that when $V_{CM} = 0$, a situation in which the bacteria do not move at all, the 2D to 3D transition occurs at the first division, that is as soon as $N_c > 1$.

It should be noted that here, just as $V_{CM}$ is a characteristic velocity and not the mean speed of the bacteria (see main text, **Figure 3A**), that the characteristic area $a(t)$ accessible to the bacteria in the colony at time $t$ is not necessarily equal to the whole colony area: first because of their finite center-of-mass velocity $V_{CM}$; secondly, because the local density may restrict their movement and the accessible surface. This effect is difficult to quantify because the fluctuations of density inside the colony area may be, depending on $V_c$, much greater than the ones encountered in the case of the Brownian diffusion of particles. Indeed, some bacteria remain static while others explore the surface extensively (main text, **Figure 3E**). Another reason is that upon division, the two daughter cells are touching and there is hence a systematic fluctuation of density upon division. Therefore the area accessible for bacteria is rather an effective measure, which cannot be directly derived from microscopic diffusion processes only. While other expressions could be used, this one is the simplest that can be proposed and matches our experimental data sufficiently well. One justification is that the underlying assumption that the velocity of bacteria is not affected by the local density (retaining the linear scaling of $a(t)$ with $V_{CM}$) is justified in the assessed situation where groups of closed-packed bacteria never exceed 5–8 cells before the 2D to 3D transition occurs. However, as a comparison, sub- and super-linear scalings will be compared with experimental data in the next section.

To further analyze the microscopic meaning of $\gamma$, we note that it is the inverse of a velocity and is related to the compactness of the colony, with higher values indicating a sparser distribution of bacteria with a lesser probability that growing/twitching bacteria will encounter several others and move to 3D because of local crowding. However it is misleading to compare it to values that could be derived from random walks with persistence because of the above-mentioned discrepancy between the colony area and the area accessible to bacteria for further spreading. Relating $\gamma$ to experimentally measured quantities on the cell movement would require a detailed analysis of the cell density fluctuations on the surface which is beyond the scope of this paper.

## B. Comparison of experimental data with the model

All available data from which characteristic velocities were extracted (main text, *Figure 3B*) were analysed and included, with the exception of one data point on glass due to the presence of an air bubble on the surface before the onset of the 2D→3D transition. The characteristic velocities for each experiment and each rigidity were taken from *Figure 3B*. The characteristic number of bacteria $N_c$ per colony was estimated as follows.

- First, for low- and medium-ridigity surfaces (2.7 kPa, 6.1 kPa and 18.5 kPa), colony formation from isolated bacteria was monitored over time until the 2D to 3D transition occurs. The number of bacteria on the surface stemming from the initial isolated bacteria were then counted, and the count for all the colonies were averaged to calculate $N_c$. In addition, the average number of colonies forming in the observed area up to that point was also measured.

- For higher-rigidity surfaces, the movement of bacteria is too large to keep track of all bacteria stemming from the same progenitor as they mix or leave the field of view, while others are incoming. As a result, $N_c$ was calculated by counting the total number of bacteria in the field of view at the time of the onset of the 2D to 3D transition, and dividing this number by the estimated number of colonies as measured on low-rigidity surfaces. It should be noted that in this case, the simple model presented above is not valid as it considers only one isolated colony, and can be expected to yield overestimated values of $N_c$. Furthermore, our evaluation method of the number of colonies in the field of view may be prone to error so we used a "blind" evaluation procedure performed before the count of bacteria in the field of view, to avoid possible biases. A change of 1 (compared to a mean value around 4) in the number of colonies used to normalise the total number of bacteria provides a good estimate of the error bars on each individual data point, and is comparable to the spread of the data points (see *Appendix 2—figure 1*). When several surfaces of the same rigidity have been measured in one experiment, the different $N_c$ values are averaged.

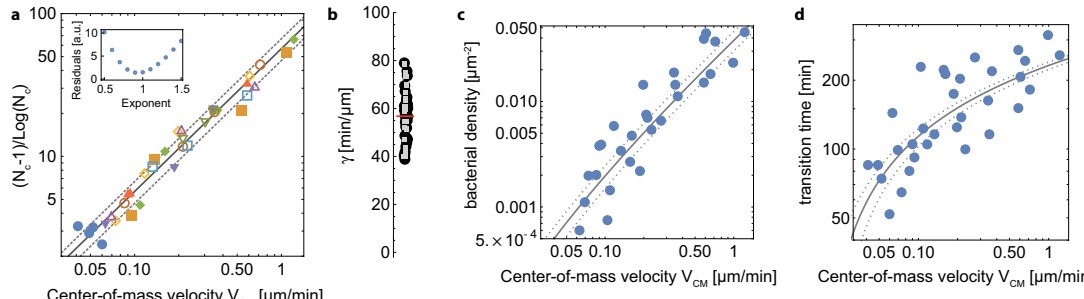

**Appendix 2—figure 1.** Experimental results and kinetic model for the transtion to 3D microcolonies. (**a**) Comparison between the experimental data (markers) and the kinetic model (lines). Blue dots are data obtained with the pili-deficient mutant pilA: Tn5. Inset, residual of the fit of all the experimental data points as a function of the exponent value for $V_{CM}$, indicating that the best fit is achieved for a value of or close to 1. (**b**) $\gamma_{exp}$ values for all data points (black disks) and their average (red line). Gray squares are data points with $v_0 < 2$ μm.min$^{-1}$, showing a similar distribution and thus ruling out a significant bias at high velocities. (**c**) Same dataset as in a but the surface density at the transition is plotted; (**d**) same dataset as in a but the time at the transition is plotted. The same model is used to describe the data, but converted into the proper quantities.

The above cited procedure produced one doublet ($V_{CM}$, $N_c$) for each rigidity of each experiment.

To match our simple model with the experimental data, *Equation A27* can be used to calculate $N_c$ as a function of $V_{CM}$ for a given value of $\gamma$. However, a direct fit of the experimental data is difficult as there is no analytical solution to *Equation A27*. Instead, an experimental value of $\gamma$ was calculated from each experimental point using the expression

$$\gamma_{exp} = \frac{N_c - 1}{V_{CM} \ln(N_c)} \tag{A.30}$$

An average experimental value is then calculated, along with a standard deviation, $\langle \gamma_{exp} \rangle = 56.8$ min. $\mu m^{-1}$ and $\delta\gamma = 11.2$ min.$\mu m^{-1}$.

*Appendix 2—figure 1a* shows the experimental data $(N_c - 1)/Log(N_c)$ as a function of $V_{CM}$ (each marker corresponds to a different experiment), and the corresponding theoretical straight lines with slopes $\langle \gamma_{exp} \rangle$ (solid line), and $\langle \gamma_{exp} \rangle$ +/- 1 standard deviation (dotted lines). To assess the deviation from the curve at high velocities, $\langle \gamma_{exp} \rangle$ was also calculated from all data points with $v_0 < 2$ $\mu m.min^{-1}$ but the change in the value is minimal (58.2 min.$\mu m^{-1}$ instead of 56.8 min.$\mu m^{-1}$, see *Appendix 2—figure 1b*).

Our strongest assumption in this modelling is the expression of $a(t)$ as a function of $V_{CM}$ [*Equation A26*]: an obvious a posteriori evidence for its correctness is that the derived equation fits our data well over more than one decade in velocity. To strengthen our point, however, we have also calculated similar curves using an exponent for $V_{CM}$ ranging from 0.5 to 1.5 (steps of 0.1, *Appendix 2—figure 1a*, inset): the comparison with experimental data indicates that reasonable agreement is only obtained for exponent values between 0.8 and 1.1, at most.

Finally we would like to point out that the data do not collapse as well when plotting the density of bacteria, or the time of the 2D→3D transition (*Appendix 2—figure 1c and d*). A likely explanation is that the initial number of bacteria on the surface varies between different datasets, a bias that is cancelled when plotting the number of bacteria instead of the density or the time at the onset of the transition. The same model is used with the same average parameter and spread, but converted into the proper quantities: for the density, the curves in *Appendix 2—figure 1a* are multiplied by the average number of colonies per observed area (3 colonies), and divided by the image area (26,121 $\mu m^2$); for the transition time, the logarithm of the number of bacteria per colony at the transition is multiplied by the typical growth time of the number of bacteria on the surface (~40 min). This time incorporates both the division time (~30 min) and the departure of a fraction of the bacteria from the surface.

