## [Editor Report]

This fundamental research significantly enhances our comprehension of the influence of substrate physical properties during the initial stages of biofilm development. By integrating microfluidics, single-cell motility, and modeling, the study provides compelling proof that mechanical interactions between the substrate and Type-IV pili drive these phenomena. This work is likely to attract a wide range of readers interested in micro-communities, their structure, and ecology.

---

## [Decision Letter]

**Decision letter after peer review:**

Thank you for submitting your article "Substrate stiffness impacts early biofilm formation by modulating *Pseudomonas aeruginosa* twitching motility" for consideration by *eLife*. Your article has been reviewed by 2 peer reviewers, and the evaluation has been overseen by Tam Mignot as the Reviewing Editor and Arturo Casadevall as the Senior Editor. The reviewers have opted to remain anonymous.

Essential revisions:

The authors conclude that motility changes are not due to physiological changes resulting from surface sensing, but rather that mechanical properties of the substrate are responsible for modulating motility differences. However, this conclusion is derived partly from the use of a chpA mutant, which the authors' data demonstrate does not exhibit differences in motility compared to WT. These data are very surprising given that several published studies demonstrate a defect in both pilus synthesis and twitching motility in PilChp mutants (including chpA). It is unclear what the differences are between the presented study and the published literature leading to the disparity in these results.

There are a number of phenotypes linked to the chpA mutant strain: are the authors observing them with the strain they used? What actually is the level of pili in the strain they manipulate? In addition, to confirm this very surprising result, it would be important to repeat it with a pilG mutant (PilG regulation by ChpA phospho-transfer drives much of the PilChp signaling pathway, so this would be a nice way to validate the surprising results shown in figure S6). This is a critical control considering the kinetic modeling and the conclusion that this phenomenon is purely mechanical is based on this result. Should the authors obtain similar results for a pilG mutant, it would be important to incorporate some discussion about what may be leading to the observed differences.

Other revisions:

– The authors conclude that twitching motility plays a key role in the rigidity modulation of microcolony formation by PA on soft elastic substrates based on data shown in supplemental figure 4. It would be nice to see confirmation that the colony morphology does not change in T4P mutants on the softest substrates (2.7 kPa pads) as a control.

– The jump in timescale needs to be better explained. Local speeds of roughly 1 um/s turn into effective speeds of um/min when one looks at a frame rate of one per minute. This might be just a reflection of the random nature of the motility but should be better explained in the text and the model. On a general note, one of the difficulties in navigating the paper as it stands is the definition of many parameters in a global manner as fits from derived equations whose assumptions are not always fully validated. For instance, Equation (1) assumes no new addition because of the flushing of the channel with the clean medium. Yet the first peak of residence time on 2.7 kPa gels is around 5 minutes per Figure S7 whereas the calculation of Vg is done over 100 minutes which should leave plenty of time for detachment and reattachment of bacteria upstream of the recording field of view. Similarly, the definition of Vcm is not easy to follow or apprehend. Is it that the general averages of the velocities are too noisy?

– The subtraction of the average velocity of the pilA mutant directly in Equation 10, could be warranted, but it would also require a little bit more explanation. All in all, a better effort in the explanation of the pros and cons of the data analysis choices will go a long way in making the article readily understandable for all. Presenting other more straightforward ways of extracting the different parameters (direct averages, choice of cut-offs…) will be helpful even in only supplementals. Another instance of assumptions that could also suffer some outliers is the assumption on line 141 that bacteria move along their long axis as it has been shown that in liquid, they can move perpendicular to it (Gibiansky et al. Science 330(6001):197).

– While the simple kinetic model presented does encapsulate many of the aspects of the data in an understandable way, some of the assumptions should be discussed further. The assumption that pili only binds with its tips is reasonable but it is very strong. While this assumption allows many simplifications in the model, type IV pili can potentially bind throughout their length, and as they can be microns in length, so can the binding region. The Koch et al. 2021b does go over the reasoning but having a small discussion earlier in the paper would be great.

– The formation of biofilms in a constant flow channel is a well characterized and common technique, yet it would be important to mention that it might be a condition quite remote to conditions found by bacteria in their common environment where the renewal of nutrients and environment might not be controlled. A small discussion around this theme might be nice.

– It could be interesting to discuss the rather large errors on different estimates of the parameters found (Vmax, E0 particularly).

- In Figure 1 and in a few figures: the 4 stiffnesses used (2.7, 18.5, 65, and 84 kPa) are not always all presented and sometimes in a non-matching set, as in Figure 1. That would be great if this could be corrected.

Text:

– The authors refer to the clusters of single cells measured in this study as colonies; this is a bit confusing since colonies generally refer to macroscopic scale (visible to the naked eye) colonies on plates. I suggest changing instances of "colony" to "microcolony" for clarity throughout.

– There is a typo in the figure legend of supplemental figure 4 and main text figure 4; it should read PAO1 pilA::Tn5 (two colons instead of one).

– The authors have a "data not shown" statement in the legend of supplemental figure 8; they should include the data (can be in the same figure, but should be shown instead of just referenced).

– Dephine instead of Delphine in the authors' names line 5.

– 85 kPa instead of 84 on page 6 of the supplemental in the legend.

*Reviewer #1 (Recommendations for the authors):*

– The authors conclude that twitching motility plays a key role in the rigidity modulation of microcolony formation by PA on soft elastic substrates based on data shown in supplemental figure 4. It would be nice to see confirmation that the colony morphology does not change in T4P mutants on the softest substrates (2.7 kPa pads) as a control.

– The authors demonstrate in supplemental figure 6 that chpA mutants have similar Vg on stiff surfaces compared to WT (figure 2C). This result is very surprising considering PilChp mutants are known to be defective in pilus synthesis and twitching motility (Bertrand et al. 2010), yet pili are presumably driving the motility seen here. To confirm this very surprising result, I would like to see it repeated with a pilG mutant (PilG regulation by ChpA phospho-transfer drives much of the PilChp signaling pathway, so this would be a nice way to validate the surprising results shown in figure S6). This is a critical control considering the kinetic modeling and the conclusion that this phenomenon is purely mechanical is based on this result. Should the authors obtain similar results for a pilG mutant, it would be nice if they incorporate some discussion about what may be leading to the observed differences.

– The authors use "typical pilus retraction speed" of 1 um/min, but this has been measured in PA to be 0.5 um/min (Skerker and Berg 2001). Would this value change result in differences in the authors' model?

*Reviewer #2 (Recommendations for the authors):*

As a follow-up on the point 1 in the Public Review: The subtraction of the average velocity of the pilA mutant directly in Equation 10, while maybe warranted, would require a little bit more explanation. All in all, a better effort in the explanation of the pros and cons of the data analysis choices will go a long way in making the article readily understandable for all. Even presenting other more straightforward ways of extracting the different parameters (direct averages, choice of cut-offs…) will be helpful even in only supplementals. Another instance of assumptions that could also suffer some outliers is the assumption on line 141 that bacteria move along their long axis as it has been shown that in liquid they can move perpendicular to it (Gibiansky et al. Science 330(6001):197).

– The formation of biofilms in a constant flow channel is a well characterized and common technique, yet it would be important to mention that it might be a condition quite remote to conditions found by bacteria in their common environment where the renewal of nutrients and environment might not be controlled. A small discussion around this theme might be nice.

– It could be interesting to discuss the rather large errors on different estimates of the parameters found (Vmax, E0 particularly).

– In Figure 1 and in a few figures: the 4 stiffnesses used (2.7, 18.5, 65, and 84 kPa) are not always all presented and sometimes in a non-matching set, as in Figure 1. That would be great if this could be corrected.

---

## [Author Response]

Essential revisions:The authors conclude that motility changes are not due to physiological changes resulting from surface sensing, but rather that mechanical properties of the substrate are responsible for modulating motility differences. However, this conclusion is derived partly from the use of a chpA mutant, which the authors' data demonstrate does not exhibit differences in motility compared to WT. These data are very surprising given that several published studies demonstrate a defect in both pilus synthesis and twitching motility in PilChp mutants (including chpA). It is unclear what the differences are between the presented study and the published literature leading to the disparity in these results.There are a number of phenotypes linked to the chpA mutant strain: are the authors observing them with the strain they used? What actually is the level of pili in the strain they manipulate? In addition, to confirm this very surprising result, it would be important to repeat it with a pilG mutant (PilG regulation by ChpA phospho-transfer drives much of the PilChp signaling pathway, so this would be a nice way to validate the surprising results shown in figure S6). This is a critical control considering the kinetic modeling and the conclusion that this phenomenon is purely mechanical is based on this result. Should the authors obtain similar results for a pilG mutant, it would be important to incorporate some discussion about what may be leading to the observed differences.

Indeed, it can be considered surprising that we observe such a high motility in a *chpA* mutant, and we thank the referee for pointing this. Most studies in the litterature report a decreased twitching for chpA mutants (see for instance [1-3]), although with different amplitudes. We have further checked the twitching of our mutant strain via macroscopic plate assays (using the method described by Turnbull and Whitchurch [4]), which confirmed that the strain is capable of twitching motility (Author response image 1). The mutant that we have used was obtained from the PAO1 transposon library, and checked by PCR upon reception using adequate primers. We thus know that the transposon is indeed present in the *chpA* gene. However, the library website indicates that the position of the insert is very close to the end of the gene sequence (PA0413, position in the ORF is 6598 out of 7419 bases), which could result in a protein that remains functional. At that stage this is our only hypothesis to explain these surprising result. To avoid any confusion we have decided to remove this piece of data from the manuscript.

**Author response image 1. sa2fig1:** Macroscopic twitching assay (A) Size of the twitching zone measured after 24h at 37*^o^*C (B) Typical plates obtained with WT and *chpA* (imaged 24h later, kept at RT). Black marks are 1cm.

Following referee 1’s suggestion, to test the behavior of bacteria impaired in pili-mediated mechanosensing, we did carry out experiments using a pilG mutant (PAO1, pilG::Tn5). However this strain clearly exhibited very low twitching motility (as confirmed by plate assays). Maybe more surprisingly, at short times in flow channels we failed to observe any significant attachment on PAA with this strain, and thus no twitching could be quantified. Either the pilG mutant is adhesiondeficient (which would be a new result), or this specific strain had further mutations and behaved aberrantly. In any case, since most mutants modified in the PilChp machinery are also impaired in twitching motility, it makes them bad candidates to use in our experiments.

Instead, we thus decided to use a strain mutated in the diguanylate cyclase sadC, which operates downstream of the PilChp system to regulate c-di-GMP level. Our results show that although it is slightly less motile than the WT, this mutant retains a stiffness-dependent motility, suggesting that pili-induced c-di-GMP regulation is not involved in the twitching modulation that we report on soft substrates. We believe this approach is also more ”logical” since we now explore in the paper two mutants for the two diguanylate cyclases (wspR and sadC) know to regulate mechanosensing in PAO1 [5].

We are well-aware that these controls are not exhaustive, and cannot be used to rule out the possibility of a regulatory response to surface attachment that could modify twitching. However, it was never our intention to strictly discard this possibility in the manuscript. We simply infer that mechanical interactions between adhering bacteria and the substrate are the primary ingredient that leads to the colony phenotypes that we observe. This has now been amply clarified in the manuscript.

[1] Bertrand JJ, West, JT, Engel, JN (2009). Genetic Analysis of the Regulation of Type IV Pilus Function by the Chp Chemosensory System of *Pseudomonas aeruginosa*. Journal of Bacteriology, 192(4), 994?1010. doi:10.1128/jb.01390-09. [2] Silversmith, R. E., Wang, B., Fulcher, N. B., Wolfgang, M. C., Bourret, R. B. (2016). Phosphoryl Group Flow within the *Pseudomonas aeruginosa* Pil-Chp Chemosensory System. Journal of Biological Chemistry, 291(34), 17677-17691. doi:10.1074/jbc.m116.737528. [3] Khn MJ et al. (2021) Mechanotaxis directs *Pseudomonas aeruginosa* twitching motility. PNAS, 118(30) e2101759118. [4] Turnbull, L. and Whitchurch, C. B. (2014). Motility Assay: Twitching Motility. *Pseudomonas* Methods and Protocols, 73?86. doi:10.1007/978-1-4939-0473. [5] Randall TE et al. (2022) Sensory Perception in Bacterial Cyclic Diguanylate Signal Transduction. Journal of Bacteriology, 204(2) doi:10.1128/jb.00433-21

Other revisions:– The authors conclude that twitching motility plays a key role in the rigidity modulation of microcolony formation by PA on soft elastic substrates based on data shown in supplemental figure 4. It would be nice to see confirmation that the colony morphology does not change in T4P mutants on the softest substrates (2.7 kPa pads) as a control.

The corresponding image has been added to Figure 1—figure supplement 4.

– The jump in timescale needs to be better explained. Local speeds of roughly 1 um/s turn into effective speeds of um/min when one looks at a frame rate of one per minute. This might be just a reflection of the random nature of the motility but should be better explained in the text and the model. On a general note, one of the difficulties in navigating the paper as it stands is the definition of many parameters in a global manner as fits from derived equations whose assumptions are not always fully validated. For instance, Equation (1) assumes no new addition because of the flushing of the channel with the clean medium. Yet the first peak of residence time on 2.7 kPa gels is around 5 minutes per Figure S7 whereas the calculation of Vg is done over 100 minutes which should leave plenty of time for detachment and reattachment of bacteria upstream of the recording field of view. Similarly, the definition of Vcm is not easy to follow or apprehend. Is it that the general averages of the velocities are too noisy?

A) Indeed, it is surprising that local (load-free) velocities of pilus retraction of1µm.s^−1^ lead only to effective bacterial velocities of 1µm.min^−1^, when measuring bacterial displacements on a time scale of minutes. The reason for this behavior is two fold. First, during each cycle of pilus extension and retraction (≈ 5s for a 5µm long pilus, consistent with Tala et al. 2019), the pilus is only attached a` short fraction of time τ=(koff0)−1≈ 1s to the substrate, which naturally reduces *k*0 the local pilus speed by a factor of koff0kp=5 for time scales longer than the pilus life cycle. Second, the pilus retraction speed slows down in a load dependent manner, reducing the local speed and consequently the effective speed even further by the factor FRFR+1≈10 (here FRFR describes the effect of bacterial friction on the tension in the pilus, with *F_R_* = 100pN and *F_B_* = 1nN for the high friction case). Both effects are nicely captured in the maximum effective speed *V*_max_ a bacterium can reach on an infinitely rigid substrate [see Equation. (8) in the main text].

We agree with the referee that the reduction of the high microscopic speed to a low effective macroscopic speed requires a more thorough explanation. We have included the above reasoning after Equation. (9) in the main text, where we discuss the model and the resulting fit parameters *V*_max_ and *E*_0_.

B) The referee is correct, in principle bacterial detachment upstream of the fieldof view and subsequent reattachment in the field of view (or migration into the field of view) is possible, and could potentially flaw the velocity analysis. However, this is not what we see in experiments. While bacterial tracking does evidence some bacterial detachment, early on bacterial attachment is a rare event (if happening at all) and cannot account for the increase in cumulative explored area S(t) [see Equation. (2)].

Two reasons could explain this observation:

– Initial surface coverage values are extremely low, with typically 2 to 4 bacteria per field of view (about 160x160 micrometer square). In addition bacteria are only present on surfaces inside the flow cell, since the upstream tubing is replaced right before starting image acquisition (a precaution we took on purpose, to avoid having too many bacteria in the flowing medium). Hence for the most part of the 100-min window that we use to determine *V_g_*, the actual number of detaching bacteria is small and results in a very low concentration of bacteria in the flow.

– These flowing bacteria might have a low adhesion probability. Consistent with results that suggest an asymmetric division event on the surface, it is possible that detaching bacteria have a poor attachment probability on the substrate, due to their specific surface properties.

C) We agree with the referee that the paper might have been hard to navigate. We have now re-written some parts, in particular the sub-section about analysis of individual bacterial displacements and relevant Figure 3, in order to improve readability. Specifically, we acknowledge the fact that the calculation of *V_g_* does rely on a few discussable assumptions (namely, no significant bacterial attachment, and directional motility along the main bacterial axis). However, the determination of *V_C_* (previously *V_CM_*) via tracking of individual bacteria, which is more time-consuming than the “explored surface area” approach, does not require any such assumption, and provides motility values very similar to the ones obtained when calculating *V_g_*. This is now stressed in the manuscript, and we believe this responds to most of the referee’s worries about unvalidated assumptions.

Similarly, we have clarified the meaning of *V_C_*. Basically, the referee is right, general averages of velocities are noisy (as highlighted Figure S9B, standard deviation for mean track velocities is typically about 50% of the determined values). We initially performed individual cell tracking in order to explore the heterogeneity of the bacterial population, as well as the temporal c” naturally” as a way to characterize the population motility. We have now added extensive explanations about the analysis of the distributions of displacement steps (in Materials and methods), including numerical calculations of theoretical distributions obtained by adding two incoherent displacement fields corresponding to active and passive displacements (Figure 3figure supplement 3). Doing so evidences the fact that *V_C_* directly corresponds to bacterial active velocity, and is not significantly affected by passive displacements (which result in a velocity *V_C_*_,*p*_ determined with the pilA mutant << *V_C_* values). Strikingly, although they result from entirely different analyses, the consistency between *V_C_* and *V_g_* values is quite remarkable.

– The subtraction of the average velocity of the pilA mutant directly in Equation 10, could be warranted, but it would also require a little bit more explanation. All in all, a better effort in the explanation of the pros and cons of the data analysis choices will go a long way in making the article readily understandable for all. Presenting other more straightforward ways of extracting the different parameters (direct averages, choice of cut-offs…) will be helpful even in only supplementals. Another instance of assumptions that could also suffer some outliers is the assumption on line 141 that bacteria move along their long axis as it has been shown that in liquid, they can move perpendicular to it (Gibiansky et al. Science 330(6001):197).

A) We have added a detailed description of the velocity data analysis in the Materials and methods section (see also point 2C). Our model distributions of displacement steps show that one does not need to substract pilA value to the values obtained for the WT. *V_C_* values are thus slightly modified in this reviewed manuscript, which only marginally changes Figure 3D.

B) Considering motion along various directions w.r.t. the bacterial axis wouldslightly modify the values of *V_g_* quantitatively, albeit velocities obtained for all rigidities would change by about the same factor. For example, using a characteristic value for bacterial dimension of 1bwb (to account for non-directional motion) would decrease the measured velocities by a factor 1bwb≈3 assuming bacterial dimensions of 1µm×3µm. As stated earlier, this simple analysis only aims at providing a qualitative readout of variations in the whole population motility. It relies on a number of assumptions, but the more detailed tracking analysis is a more accurate and thorough way to investigate bacterial velocity.

– While the simple kinetic model presented does encapsulate many of the aspects of the data in an understandable way, some of the assumptions should be discussed further. The assumption that pili only binds with its tips is reasonable but it is very strong. While this assumption allows many simplifications in the model, type IV pili can potentially bind throughout their length, and as they can be microns in length, so can the binding region. The Koch et al. 2021b does go over the reasoning but having a small discussion earlier in the paper would be great.

We included a discussion of the simple assumption that the pilus attaches as its extremity in the main text when introducing the minimal kinetic model (l. 184) and into appendix 1.

– The formation of biofilms in a constant flow channel is a well characterized and common technique, yet it would be important to mention that it might be a condition quite remote to conditions found by bacteria in their common environment where the renewal of nutrients and environment might not be controlled. A small discussion around this theme might be nice.

In our experimental approach we initially followed the idea to study the influence of a single environmental property at a time (here substrate stiffness), trying to maintain otherwise permanent conditions. However, the referee raises an interesting point, since outside of the lab *Pseudomonas aeruginosa* can invade environments where it is exposed to little to no shear (lung alveolae, or wound surfaces are two examples). In parallel to this study, we have now pursued work where we observe microcolony formation in the absence of flow or medium renewal, in wells. We have added a figure (Figure 1—figure supplement 3) showing that there is a clear difference in bacterial organization in that case as well, with bacteria forming denser colonies on softer PAA, similar to what is observed under flow. There is a small discussion around this in the main text (starting l.111).

– It could be interesting to discuss the rather large errors on different estimates of the parameters found (Vmax, E0 particularly).

The error estimates were calculated directly from the co-variance matrix of the fit function and the variance of residuals (chi-squared sum divided by the number of degrees of freedom) of the non-linear fit in gnuplot (version 5.2.8 for Mac OSX). They are reflective of the wide scattering of measured velocities between different experiments (albeit the trend in velocity increase for increasing rigidities is the same for all individual experiments, see e.g Figure 3B). We have added a remark (l.211) on the origin of the big error estimates in the main text after Equation. 9 where we discuss the fitted parameters w.r.t. to measured microscopic parameters of the twitching process.

- In Figure 1 and in a few figures: the 4 stiffnesses used (2.7, 18.5, 65, and 84 kPa) are not always all presented and sometimes in a non-matching set, as in Figure 1. That would be great if this could be corrected.

The geometry of the channels used in our experiments, and the size of hydrogels are such that we were only able to observe 3 hydrogels in a given experiment, in a 5-cm long channel. For that reason, and because of the variability in measurements, on figures that show selected images we use images acquired on the same day, and cannot display all 4 tested rigidities. Unfortunately we cannot easily correct this. The main experimental limitation is the small volume of hydrogel premix (3 µ*l* in our case to obtain a 12mm, 35 microns-thick hydrogel patch) that needs to be deposited on the substrate, that prevented us from downscaling the size of hydrogels in order to have more rigidity values in a single 5 cm long channel (we tried, but hydrogel premix would evaporate very rapidly when the droplet volume was too small, which impacted the spatial homogeneity of the final gel).

Text:– The authors refer to the clusters of single cells measured in this study as colonies; this is a bit confusing since colonies generally refer to macroscopic scale (visible to the naked eye) colonies on plates. I suggest changing instances of "colony" to "microcolony" for clarity throughout.– There is a typo in the figure legend of supplemental figure 4 and main text figure 4; it should read PAO1 pilA::Tn5 (two colons instead of one).– The authors have a "data not shown" statement in the legend of supplemental figure 8; they should include the data (can be in the same figure, but should be shown instead of just referenced).– Dephine instead of Delphine in the authors' names line 5.– 85 kPa instead of 84 on page 6 of the supplemental in the legend.

These mistakes have all been corrected. Figure S8 (now Figure 3—figure supplement 1) has been modified as suggested. We thank the referees for their careful reading of the manuscript.